# Meisosomes, folded membrane microdomains between the apical extracellular matrix and epidermis

Dina Aggad[1†], Nicolas Brouilly[2†], Shizue Omi[1†], Clara Luise Essmann[3,4], Benoit Dehapiot[2], Cathy Savage-Dunn[5], Fabrice Richard[2], Chantal Cazevieille[6], Kristin A Politi[7], David H Hall[7], Remy Pujol[6], Nathalie Pujol[1]*

[1]Aix Marseille Univ, INSERM, CNRS, CIML, Turing Centre for Living Systems, Marseille, France; [2]Aix Marseille Université, CNRS, IBDM, Turing Centre for Living Systems, Marseille, France; [3]Department of Computer Science, University College London, London, United Kingdom; [4]Bio3/Bioinformatics and Molecular Genetics, Albert-Ludwigs-University, Freiburg, Germany; [5]Department of Biology, Queens College and the Graduate Center, CUNY, Flushing, United States; [6]INM, Institut des Neurosciences de Montpellier, Plateau de microscopie électronique, INSERM, Université de Montpellier, Montpellier, France; [7]Department of Neuroscience, Albert Einstein College of Medicine, New York, United States

*For correspondence:
pujol@ciml.univ-mrs.fr

[†]These authors contributed equally to this work

Competing interest: The authors declare that no competing interests exist.

**Abstract** Apical extracellular matrices (aECMs) form a physical barrier to the environment. In *Caenorhabditis elegans*, the epidermal aECM, the cuticle, is composed mainly of different types of collagen, associated in circumferential ridges separated by furrows. Here, we show that in mutants lacking furrows, the normal intimate connection between the epidermis and the cuticle is lost, specifically at the lateral epidermis, where, in contrast to the dorsal and ventral epidermis, there are no hemidesmosomes. At the ultrastructural level, there is a profound alteration of structures that we term 'meisosomes,' in reference to eisosomes in yeast. We show that meisosomes are composed of stacked parallel folds of the epidermal plasma membrane, alternately filled with cuticle. We propose that just as hemidesmosomes connect the dorsal and ventral epidermis, above the muscles, to the cuticle, meisosomes connect the lateral epidermis to it. Moreover, furrow mutants present marked modifications of the biomechanical properties of their skin and exhibit a constitutive damage response in the epidermis. As meisosomes co-localise to macrodomains enriched in phosphatidylinositol (4,5) bisphosphate, they could conceivably act, like eisosomes, as signalling platforms, to relay tensile information from the aECM to the underlying epidermis, as part of an integrated stress response to damage.

## Editor's evaluation

This valuable work addresses the cellular mechanisms that mediate attachment of the lateral epidermis to the cuticle. The evidence supporting the role of structures called 'meisosomes' by the authors is solid, and addresses the roles of these structures in maintaining and patterning the epidermis and cuticle. The work will be of interest to developmental and cell biologists.

## Introduction

Apical extracellular matrices (aECMs) are associated with all epithelia and are essential for animal life. In *Caenorhabditis elegans*, a collagen-rich aECM covers the entire surface of the worm and is called

the cuticle. It is a complex multilayer structure that acts as an exoskeleton, to which body-wall muscles are connected via structures called hemidesmosomes that traverse the intervening epidermis (*Davies and Curtis, 2011*; *Johnstone, 2000*; *Page and Johnstone, 2007*). Specific subsets of the more than 170 collagens are enriched in the different layers of the cuticle. Some are involved in the formation of distinct structures, including the circumferential parallel furrows that cover the entire animal (*Cox and Hirsh, 1985*; *Cox et al., 1980*; *McMahon et al., 2003*; *Page and Johnstone, 2007*; *Thein et al., 2003*), and the longitudinal lateral alae. The latter have been proposed to be involved in facilitating the traction of *C. elegans* to its substrate during locomotion, although notably they are not present from the L2 through the L4 larval stages (*Cox et al., 1981*; *Katz et al., 2022*).

The cuticle also constitutes a physical barrier, protecting the underlying epidermal syncytium from biotic and abiotic stresses. When the cuticle is damaged, mechanically or through infection, the epidermis reacts, activating an immune response, reflected by the increased expression of anti-microbial peptide (AMP) genes, including those of the *nlp-29* cluster (*Belougne et al., 2020*; *Pujol et al., 2008a*; *Taffoni et al., 2020*). An elevated level of AMP gene expression is also observed in a subset of mutants that affect the cuticle, specifically those that lack furrows (*Dodd et al., 2018*; *Pujol et al., 2008b*; *Zugasti et al., 2016*). These six known furrowless mutants (*dpy-2, 3, 7, 8, 9*, and *10*) exhibit other characteristic physiological alterations, including an activation of detoxification genes, dependent on the Nrf transcription factor SKN-1, and the induction of genes required for osmolyte accumulation (*Dodd et al., 2018*).

If the pathway leading to AMP induction in the epidermis is well described (reviewed in *Martineau et al., 2021*), exactly how the epidermis senses cuticular damage remains obscure. Part of the induction seen in *dpy-10* mutants is the consequence of an increase in the levels of hydroxyphenyllactic acid (HPLA). This metabolite, derived from tyrosine by transamination and reduction, activates the G-protein coupled receptor (GPCR) DCAR-1 (*Zugasti et al., 2014*), switching on a signalling cascade that leads to AMP gene expression (*Polanowska et al., 2018*; *Zugasti et al., 2016*). What provokes elevated HPLA levels in *dpy-10* mutants is, however, currently unknown. Further, the HPLA/DCAR-1 signalling pathway only accounts for part of the elevated *nlp-29* expression seen in furrow collagen mutants (*Zugasti et al., 2014*). We have therefore proposed that a hypothetical, cuticle-associated, damage sensor exists that would control, in an as yet undefined manner, AMP gene expression. This damage sensor would also function to coordinate antimicrobial responses with the distinct detoxification and hyperosmotic responses that are simultaneously activated in furrow collagen mutants (*Dodd et al., 2018*; *Rohlfing et al., 2010*; *Wheeler and Thomas, 2006*).

In yeast, eisosomes, single invaginations of the plasma membrane underneath the aECM, the cell wall, are responsible for detecting changes in nutrient availability, but also cell wall integrity and membrane tension. They relay information primarily via BAR domain proteins to induce the appropriate responses to environmental changes (*Appadurai et al., 2020*; *Lanze et al., 2020*; *Moseley, 2018*). While nematodes lack eisosomes, the apical plasma membrane of the epidermis, which is in direct contact with the aECM (the cuticle), is characterised by localised regions of folds that can be observed by electron microscopy (*Liégeois et al., 2006*; *White et al., 1986*; *Wood, 1988*). Given their superficial similarity, we refer to these structures as meisosomes, for multifold eisosomes. In this study, we undertook a detailed ultrastructural analysis of meisosomes in adults, as well as characterising them during development, and in furrowless mutants. This mutant analysis proposed a role for meisosomes in maintaining the structural integrity of the cuticle and the epidermis, and has opened the way to future, more detailed, characterisation of their function.

## Results

### Meisosomes: Epidermal plasma membrane folds interfacing the apical ECM

The stacked organelles that we refer to as meisosomes were mentioned during early electron microscopy characterisation of *C. elegans* (*Wood, 1988*). A survey of the long transverse transmission electron microscopy (TEM) series 'N2U' from the MRC archive (*White et al., 1986*), which is of a 4–5-day-old adult hermaphrodite, found hundreds of meisosomes across the 400 odd available transverse prints in the midbody. As a first step in the detailed investigation of meisosomes, we undertook a focused TEM study to determine their 2D organisation and their 3D structure. Meisosomes are repeated folded

structures at the interface of the aECM (the cuticle) and the epidermis (*Figure 1*). They can be found in similar locales at all larval stages, predominantly in the epidermal syncytium hyp7 at the lateral, dorsal, and ventral ridges, and in the tail tip epidermal cells. They are not present on the basal side of the epidermis, nor in the seam cells, the rectal epithelia, nor in the pharynx (*Figure 1*, *Figure 1—figure supplement 1*). In adults, meisosomes typically comprised 4–10 closely apposed parallel folds of the plasma membrane, although we observed some with up to 30-folds (*Figure 1C*). Most folds formed an indentation 200–400 nm deep (*Figure 1C–G*). The folds were regularly spaced. The gap between each cytoplasmic-facing plasma membrane fold was 35 nm, 75% wider than that between the folds made from cuticle-facing plasma membrane (20 nm) (*Figure 1G*). The cytoplasmic faces of the folds were free of ribosomes but contained dense material close to the plasma membrane, separated by a thin less electron-dense area (*Figure 1G*). Meisosomes were frequently found in close proximity to mitochondria (85%, n = 355) (*Figure 1C–E*). On their apical side, some folds were found close to a furrow (*Figure 1C and E*). Although very variable in a single worm, meisosomes of similar dimensions were observed in both transverse and longitudinal sections (*Table 1*), consistent with a random orientation relative to the animal's antero-posterior axis. This random orientation was clearly visible in electron micrographs of freeze-fractured samples (*Figure 1—figure supplement 2A and B*). It contrasted with a much more regular pattern in moulting larvae, in which meisosomes were in-between the position of circumferential furrows, (*Figure 1—figure supplement 2C and D*). As described below, a similar organisation could be observed through moulting using in vivo markers. Some much smaller meisosomes, typically with only 2–4 shallow folds, were seen in the thin epidermal tissue that lies between body-wall muscles and the cuticle (dorsal and ventral epidermis, see *Figure 1A*) and that is largely devoid of cytoplasmic content (*Figure 1—figure supplement 3*).

To understand meisosomes' 3D structure, we undertook a tomographic analysis on serial 350 nm-thick sections. This approach confirmed the existence of groups of parallel folds, all found in continuity with the plasma membrane (*Figure 2*). The tomographic analysis also revealed variability in the geometry of the folds. Although most groups of folds were perpendicular to the apical surface, some were tilted. The folds had a relatively uniform depth, but were deeper at the centre of each stack. No break in the plasma membrane was observed on neither the apical nor the basal side of the meisosomes. Despite their close apposition with mitochondria, no membrane continuity was observed between meisosomes and mitochondria (*Figure 2A–D*, *Figure 2—video 1*).

In order to evaluate not just the topology, but also the distribution of these organelles, we developed a fixation protocol for serial block-face scanning electron microscopy (SBF-SEM) of *C. elegans* samples. Starting with protocols previously described (*Deerinck et al., 2010*; *Hall et al., 2012a*; *Hall et al., 2012b*), we adapted the solvents and temperatures for each contrasting agent, including lead and uranyl acetate, to maximise sample contrast (see 'Materials and methods'). We acquired series of electron micrographs of the lateral epidermis as transversal views along 12 µm. We produced a voxel-based classification within the Waikato Environment for Knowledge Analysis (WEKA in Fiji) and then used its machine learning algorithms to perform semi-automated organelle recognition. This revealed that the meisosomes were irregularly spaced at the apical surface of the lateral epidermis, with various sizes and orientations and confirmed their frequent apposition to mitochondria (*Figure 2—figure supplement 1*).

## VHA-5 is a marker of meisosomes

VHA-5, one of four α-subunits of the transmembrane V0 complex of the vacuolar ATPase (V-ATPase) (*Oka et al., 2001*; *Pujol et al., 2001*), and RAL-1, ortholog of human RALA (RAS like proto-oncogene A) (*Frische et al., 2007*), are the only known markers of meisosomes, with both proteins being also associated with multivesicular bodies (MVBs) that play a role in exosome secretion (*Hyenne et al., 2015*; *Liégeois et al., 2006*). The expression pattern of VHA-5 is the better characterised of the two proteins. A TEM/immunogold staining study showed that more than 85% of the VHA-5 signal can be attributed to meisosomes (*Liégeois et al., 2006*).

We have used several VHA-5 reporter strains, including one expressing a GFP-tagged version of VHA-5 from a CRISPR/Cas9 engineered genomic locus [KI], a single copy insertion under an epidermis promoter [Si], or classic multi-copy integrated [Is] or extrachromosomal [Ex] transgenic arrays. In all the strains, and in line with previous reports (*Liégeois et al., 2006*), we observed the same punctate fluorescence at the apical surface of the lateral epidermis, and in the ventral and dorsal ridges, from the

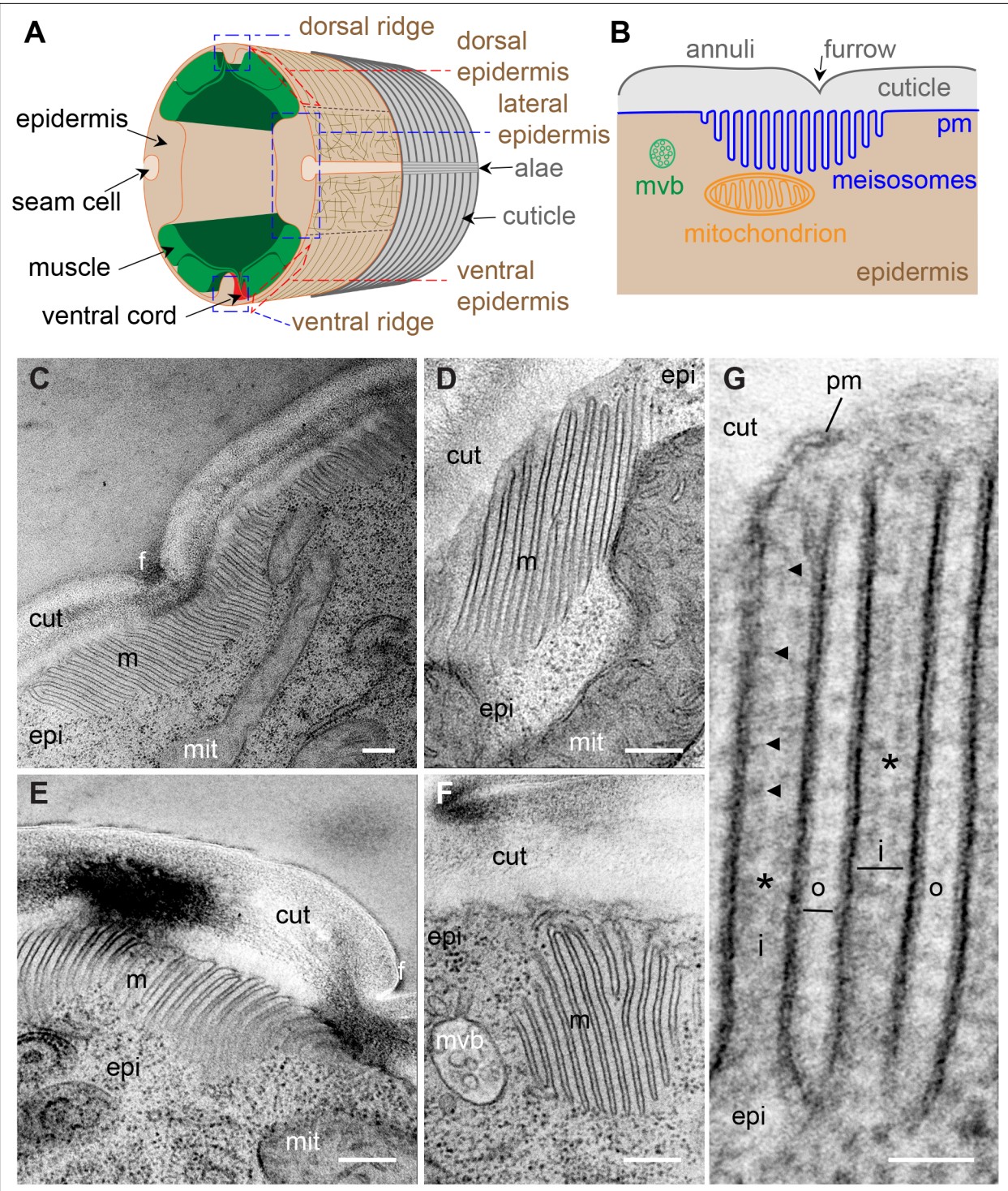

**Figure 1.** Meisosomes are membrane-folded structures on the apical side of the lateral epidermis. (**A**) Schematic view of the outer tissues of an adult *C. elegans* worm near the mid-body. The different regions of the epidermis containing meisosomes are boxed in blue (lateral, and dorsal and ventral ridges), the regions above the muscles that are extremely thin are boxed in red. (**B**) Schematic view of the connection between the cuticle and the plasma membrane (pm) of the lateral epidermis highlighting the position of meisosomes relative to multivesicular bodies (mvb) and mitochondria. (**C–G**) TEM images of longitudinal sections of young adult wild-type hermaphrodite worms reveal epidermal structures, meisosomes (m), contacting the cuticle (cut), composed of parallel plasma membrane (pm) folds. (**C–F**) Meisosomes typically comprise less than ten folds, but can have more than 30, as shown in (**C**) where the meisosome is 4 µm wide, and extends on both sides of a furrow (f). Some folds are apposed to mitochondria (mit) (**D**), can vary in size and orientation (**C–F**) and can appear not to be in direct contact with the cuticle in some EM sections (**F**); scale bar, 200 nm. (**G**) High-magnification

*Figure 1 continued on next page*

*Figure 1 continued*

view of plasma membrane (pm) folds. The 'internal' folds (i) are 35 nm wide, contain a ribosome-free cytoplasm but ladder-like banding (indicated by the black arrowheads), and are less electron dense in their middle (*). The 'outer' folds (o) on the cuticle side are 20 nm wide. Epidermis (epi); scale bar, 50 nm.

The online version of this article includes the following figure supplement(s) for figure 1:

**Figure supplement 1.** Meisosomes are present in epidermis at all development stages.

**Figure supplement 2.** Meisosomes are randomly orientated except before moulting.

**Figure supplement 3.** Smaller meisosomes can be found over the muscle quadrants.

head to the tail. As expected, it was almost completely absent from the dorsal and ventral epidermis above the body-wall muscles, and underneath the seam cells (*Figure 3A–C*). By combining a MUP-4::GFP (*Suman et al., 2019*) and a VHA-5::RFP reporter (*Liégeois et al., 2006*), we confirmed the complementary pattern in the epidermis of hemidesmosomes above the muscles, and meisosomes in the lateral epidermis and dorso/ventral ridges (*Figure 3D*). Higher resolution analysis in the lateral epidermis revealed the VHA-5-marked structures to have an irregular shape in no preferred direction in the lateral epidermis, consistent with the TEM and SBF-SEM analyses (*Figure 3E*).

When we examined worms co-expressing VHA-5::GFP and HGRS-1::mScarlet, a known marker of MVBs and the endosome degradation pathway (*Liégeois et al., 2006*; *Norris et al., 2017*; *Roudier et al., 2005*; *Serrano-Saiz et al., 2020*), we observed essentially no co-localisation. Further, the patterns of the two markers were distinct, with HGRS-1::mScarlet labelling structures that were less apical, larger and more scattered that those labelled by VHA-5::GFP (*Figure 3F*, *Figure 3— figure supplement 1A*). We also did not observe any co-localisation between VHA-5::GFP and SNX-1::mScarlet, a marker for the recycling endosomes, nor with LGG-1::mScarlet, an autophagolysosome marker (*Serrano-Saiz et al., 2020*; *Figure 3—figure supplement 1B and C*). We have previously shown that the plasma membrane of the epidermis in young adult worms contains heterogeneous macrodomains that can be revealed with a prenylated, or a pleckstrin homology domain-tagged form of GFP (GFP::CAAX and GFP::PH-PLC1δ, respectively) (*Taffoni et al., 2020*). Interestingly, in worms co-expressing VHA-5::RFP and either one of these membrane probes, we observed a high degree of co-localisation (*Figure 3G and H*, *Figure 3—figure supplement 1D–F*). This further reinforces the notion that VHA-5 is primarily a marker of subdomains of the plasma membrane.

The structures labelled with both VHA-5::GFP and CAAX::GFP or PH-PLCδ::GFP in the adult epidermis were similar in size and spatial distribution to the meisosomes reconstituted from the

**Table 1.** Quantification of the length of the meisosomes on TEM images in young adult wild-type and different collagen mutants.
The orientation of the section, transversal or longitudinal, the number of different worms observed, and the number of meisosomes analysed are reported.

| Genotype | Cut | n (worms) | n (meisosome) | Mean length (μm) | Smallest length (μm) | Longest length (μm) |
|---|---|---|---|---|---|---|
| Wild-type | Transv. | 5 | 22 | 0.71 | 0.2 | 2.6 |
| Wild-type | Longit. | 4 | 41 | 0.77 | 0.1 | 4 |
| *dpy-13(e184)* | Transv. | 3 | 31 | 0.77 | 0.3 | 4 |
| *dpy-13(e184)* | Longit. | 1 | 11 | 1.07 | 0.2 | 4 |
| *dpy-2(e8)* | Transv. | 1 | 45 | 0.39 | 0.05 | 1 |
| *dpy-2(e8)* | Longit. | 4 | 6 | 0.28 | 0.1 | 0.6 |
| *dpy-3(e8)* | Longit. | 1 | 16 | 0.49 | 0.1 | 1 |
| *dpy-7(e88)* | Transv. | 2 | 24 | 0.28 | 0.05 | 0.7 |
| *dpy-7(e88)* | Longit. | 4 | 37 | 0.37 | 0.1 | 1.5 |

The online version of this article includes the following source data for table 1:

**Source data 1.** Measurements of meisosome length on TEM images in young adult wild type and different collagen mutants.

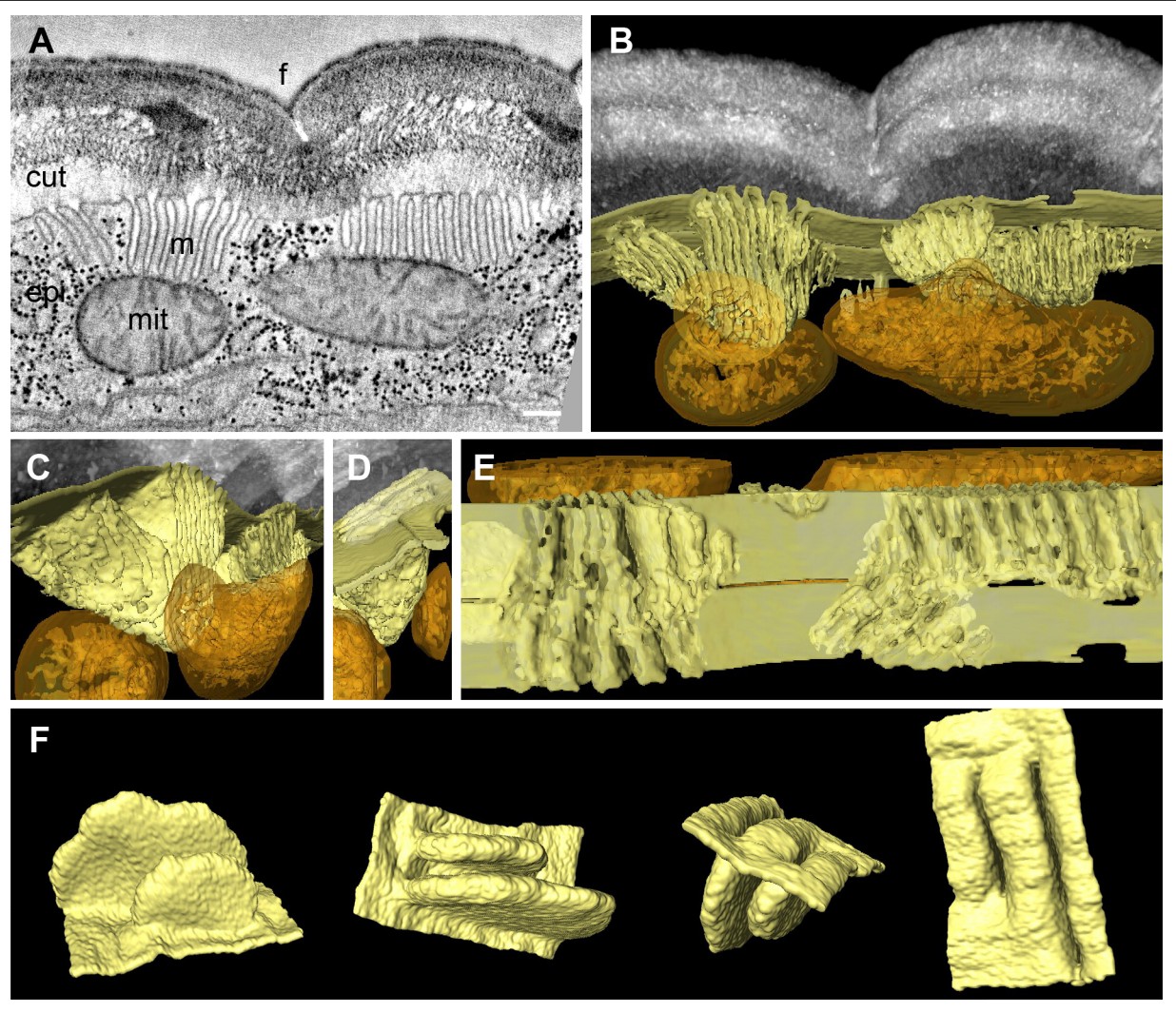

**Figure 2.** Meisosomes are formed by epidermal plasma membrane facing the apical extracellular matrix (ECM). Adjacent meisosomes in two serial thick (300 nm) sections were analysed with by electron tomography. (**A**) Selected virtual image from the serial reconstruction. (**B–E**) Segmentation of membranes and mitochondria reveal their 3D topology. Meisosomes (in yellow) are in close apposition to, but not in continuity with, mitochondria (orange) and are formed by epidermal plasma membrane folds, as observed in an *en face* view after removing the cuticle in silico (**E**). (**F**) Two folds were extracted and manually filled for a schematic view. Cuticle (cut), epidermis (epi), furrow (f), mitochondria (mit), and meisosomes (m); scale bar, 200 nm.

The online version of this article includes the following video and figure supplement(s) for figure 2:

**Figure supplement 1.** Meisosome distribution by serial block-face scanning electron microscopy (SBF-SEM).

**Figure 2—video 1.** Visualisation of the electron tomography and the 3D segmentation of meisosomes.

https://elifesciences.org/articles/75906/figures#fig2video1

SBF-SEM data (*Figure 2—figure supplement 1D*). To confirm that the observable fluorescence signal from VHA-5::GFP indeed primarily originated from meisosomes, we performed correlation light and electron microscopy (CLEM) using the VHA-5::GFP [Si] strain in which the strong and potentially confounding excretory canal GFP signal is absent due to the use of an heterologous epidermis-specific promoter. As we used a different fixation technique to preserve the GFP signal (*Johnson et al., 2015*) and we worked on semi-thin section, meisosomes were revealed by electron tomography. After alignment of the confocal and TEM images, we could show that the fluorescence foci matched meisosomes (*Figure 3I and J*, *Figure 3—figure supplement 2*). Together with previous CLEM observations in the excretory duct (*Kolotuev et al., 2009*), these results indicate that the VHA-5::GFP signal that we observe at the apical membrane in the epidermis corresponds to meisosomes and that VHA-5::GFP can be used in vivo as a *bona fide* meisosome marker for this study.

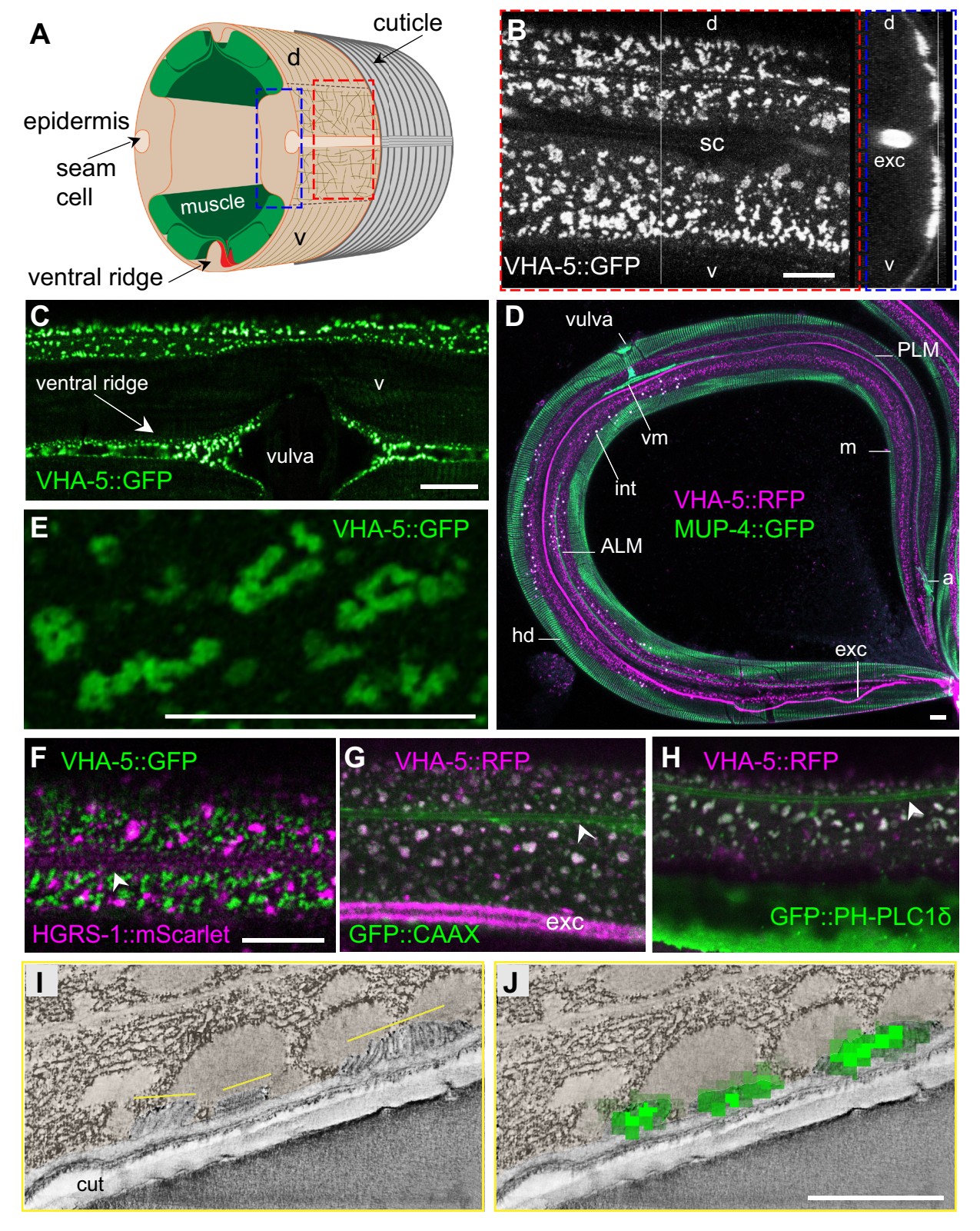

**Figure 3.** VHA-5 is a marker of plasma membrane containing meisosomes. (**A**) Schematic view of the outer tissues of an adult *C. elegans* worm near the mid-body, with the boxed *en face* view (red) or z projection (blue) for fluorescence microscopy. (**B–H**) Confocal images of young adult worms using different strains expressing either a single copy insertion of VHA-5::GFP under an epidermis promoter [SI], or a classic multi-copy integrated VHA-5::RFP [Is] or extrachromosomal VHA-5::GFP [Ex] transgenic array. (**B**) In a young adult worm, VHA-5::GFP fluorescence from a multicopy transgene [Ex] is

*Figure 3 continued on next page*

*Figure 3 continued*

observed in the lateral epidermis, as well as in the excretory canal (exc), but largely excluded from the ventral (v) and dorsal (d) regions above the muscle quadrants, known to contain hemidesmosomes, as well as the region above the seam cell (sc), as observed in *en face* (left panel), and orthogonal (right) projections of reconstructed confocal stacks. (**C**) VHA-5::GFP is also expressed in the ventral ridge, as observed in a ventral view. (**D**) High-resolution imaging on Airyscan mode revealed VHA-5::GFP from a single copy insertion [Si] to be associated with long and convoluted apical membranous structures. (**D**) Meisosomes (m), visualised using a VHA-5::RFP integrated transgene [Is] (magenta), are complementary with hemidesmosomes (hd) visualised in a MUP-4::GFP KI (green), autofluorescence, in white, is from intestinal granules. Tiled image acquired with the spectral mode z projection of six sections of 1 µm; exc, excretory canal. Attachment of different structures to the epidermis with hd are highlighted: vulval muscles (vm), ALM and PLM neurones, anal muscles (a). (**F–H**) Confocal images of young adult worms expressing VHA-5::GFP [Ex] (green) together with an HGRS-1::mScarlet marker (magenta) (**F**), a VHA-5::RFP [Is] (magenta) together with a CAAX::GFP (**G**) or PH-PLC1δ::GFP marker (green) (**H**); see associated *Figure 3—figure supplement 1* for the individual channel as well as the co-localisation quantifications. Scale bar, 10 µm. (**I–J**) Correlative light electron microscopy (CLEM) aligns the position of three meisosomes revealed by tomography (**I**) with three spots of VHA-5::GFP signal imaged by confocal imaging (**J**), see associated *Figure 3—figure supplement 2* for the detailed procedure and another example. Epidermis is pseudo-coloured in beige. Scale bar, 1 µm.

The online version of this article includes the following figure supplement(s) for figure 3:

**Figure supplement 1.** VHA-5::FP co-localise with membrane markers but not with vesicular markers in the epidermis.

**Figure supplement 2.** Correlative light electron microscopy (CLEM).

## Meisosomes align in between furrows before moulting

VHA-5 has been shown to have an essential role in alae formation and secretion of Hedgehog-related peptides through exocytosis via MVBs, but not to be involved in secretion of the collagen DPY-7, nor in meisosome morphology (*Liégeois et al., 2006*). Indeed, to date, no gene has been assigned a role in determining meisosome structure. As a path to understanding their function, we first observed their morphology during development. Consistent with previous reports (*Liégeois et al., 2007*), we observed that VHA-5::GFP aligns parallel to the actin fibres in animals entering the L4/adult moult, a stage we refer to here as 'late L4.' We refined this observation by precisely staging the worms throughout the L4 stage on the basis of vulval morphology and the shape of the lumen as previously described in *Cohen et al., 2020*; *Mok et al., 2015*; *Figures 4A and 5*. The parallel circumferential alignment of VHA-5::GFP could be observed at the beginning of the L4 stage, in L4.1 worms, but was then lost at the L4.2 stage. It reappeared progressively starting in the L4.3 stage, culminating between the L4.7 and L4.9 stages, just before the moult. This is consistent with the EM observations of meisosome alignment before moulting (*Figure 1—figure supplement 2C and D*).

As meisosomes, like the rest of the apical plasma membrane, are in direct contact with the aECM, the cuticle, we investigated the relation of meisosomes to different cuticle components. Different classes of cuticular collagen exist that form either the circumferential constricted furrows, or the cuticle in the regions between the furrows, called the annuli (*Cox and Hirsh, 1985*; *Cox et al., 1980*; *McMahon et al., 2003*; *Page and Johnstone, 2007*; *Thein et al., 2003*). As revealed with one marker of a furrow collagen, DPY-7::GFP (*Miao et al., 2020*), in combination with VHA-5::RFP, meisosomes align in between the furrows at the late L4 stage (*Figure 4B*). Notably, during the L4.7 stage, some DPY-7::GFP can be observed in small vesicles on the apical side of the epidermis, which could represent the ongoing secretion of furrow collagen at that stage. Interestingly, these vesicles do not co-localise with VHA-5::RFP (*Figure 4B*, right panel). We further show that the CAAX and PH-PLC1δ markers that co-localise with VHA-5 in young adult animals (*Figure 3G and H*) also align during the L4 stage, but other vesicular components, like the one marked by HGRS-1, SNX-1, or LGG-1 do not (*Figure 4C and D*). Thus, meisosomes, together with specific membrane subdomains, align in between the furrow before moulting.

## Furrow collagens determine the organisation of the cytoskeleton and meisosomes in L4 larvae

Before moulting, there is profound reorganisation of the cytoskeleton in the lateral epidermis. Microtubules and actin fibres align in a series of circumferential bands that are not present in adults (*Castiglioni et al., 2020*; *Costa et al., 1997*; *Taffoni et al., 2020*). Interestingly, meisosomes exhibited the same sequence of dynamic changes in alignment as microtubules and actin. After the moult, there was a concomitant loss of alignment of VHA-5, actin, and microtubules, so that in wild-type adult animals, as described above, there was no clear overall pattern to the organisation of meisosomes, actin, or microtubules (*Figure 5*).

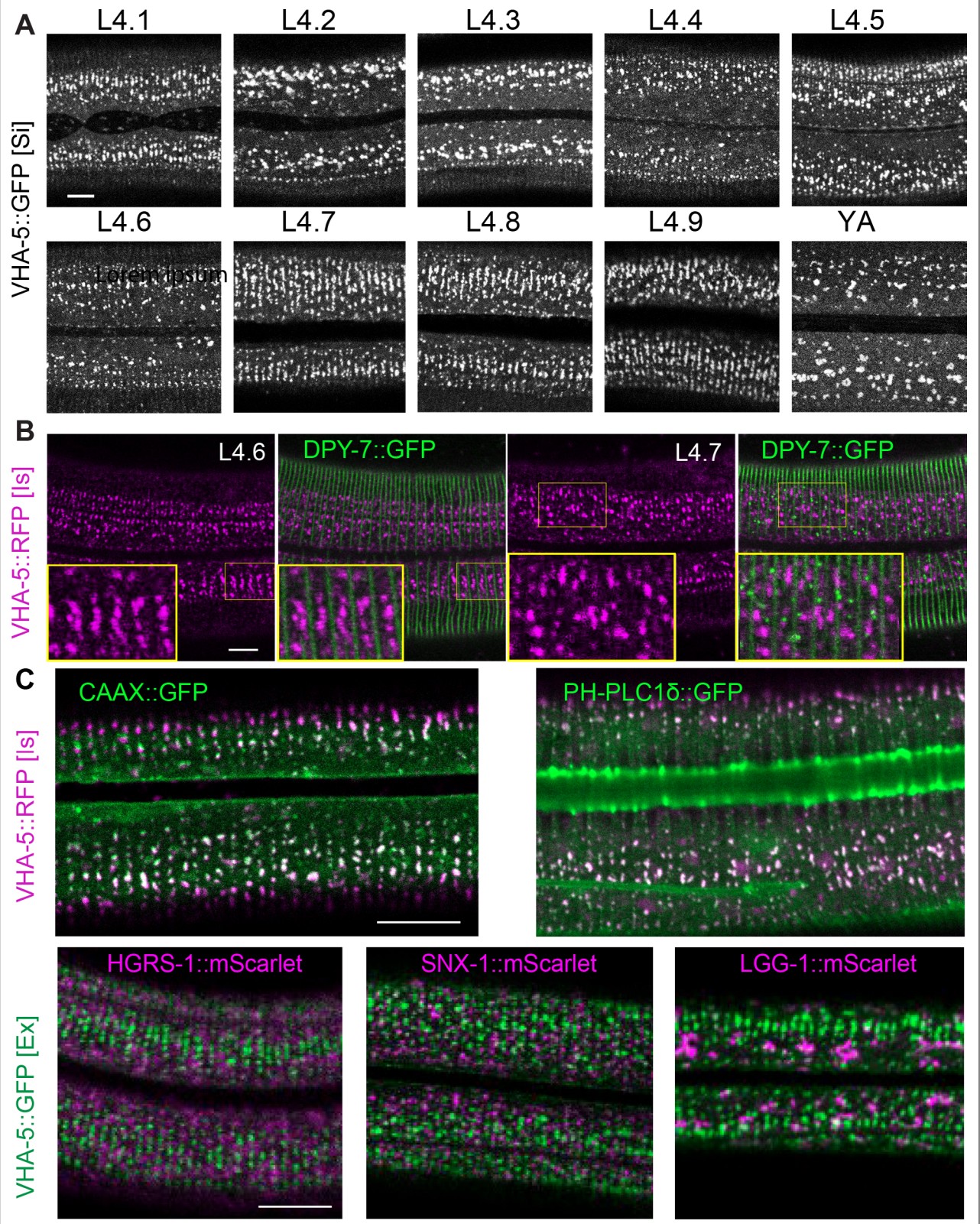

**Figure 4.** Meisosome aligned in between furrow before moulting. (**A**) Confocal images of worms expressing VHA-5::GFP [Si] from early L4 to young adult (YA) stage. To define the precise L4 stage of all observed worms, the vulva was observed and worms classified according to *Cohen et al., 2020*; *Mok et al., 2015*. (**B**) Confocal images of worms expressing both DPY-7::GFP and VHA-5::RFP, RFP channel on the left, merge channel on the right. The inserts show selected regions at a ×1.6 higher magnification. (**C**) Merged RFP and GFP channel confocal images of lateral epidermis of late L4 worms

*Figure 4 continued on next page*

*Figure 4 continued*

expressing: upper panel, a VHA-5::RFP [Is] with a CAAX::GFP (left panel) or a PH-PLC1δ::GFP (right panel); lower panel, a HGRS-1::mScarlet (left panel), SNX-1::mScarlet (middle panel) or LGG-1::mScarlet (right panel) with VHA-5::GFP [Ex]. Scale bar, 10 μm.

We then examined the consequence of knocking down the expression of one of the furrow collagen genes, *dpy-7*, on the organised VHA-5::GFP pattern of late L4s worms. Compared to the control, RNAi of *dpy-7* provoked a loss of meiosome alignment (*Figure 5*, *Figure 5—figure supplement 1*). A similar phenotype was observed in *dpy-3* mutant worms; DPY-3 is another furrow collagen (*Figure 5—figure supplement 1B*). Strikingly, this loss of furrow collagens was also associated with a disruption of the normal organised pattern of both actin fibres and microtubules from the L4.3 stage (*Figure 5*). It was previously proposed that the formation of actin fibres and microtubules in apposed circumferential bands plays an instructive role in positioning the furrows (*Costa et al., 1997*). Our results suggest, on the contrary, that furrow collagens in the cuticle govern the alignment of the underlying cytoskeleton as development progresses into the final moult. Thus, furrow collagens appear to be required to align both meiosomes and the actin-microtubule cytoskeleton.

## Abnormal meiososomes in adult furrow mutant worms

As previously mentioned, different classes of cuticular collagen exist that are expressed and form either furrows or annuli (*Figure 6A*; *Cox and Hirsh, 1985*; *Cox et al., 1980*; *McMahon et al., 2003*; *Page and Johnstone, 2007*; *Thein et al., 2003*). While mutants in these collagens all have a Dumpy (i.e. short and fat; Dpy) phenotype, only the furrow-less mutants, in contrast to mutants of annuli collagens, exhibit an increased expression of the AMP reporter *nlp-29*p::GFP (*Dodd et al., 2018*; *Pujol et al., 2008b*; *Zugasti et al., 2014*; *Zugasti et al., 2016*). This is one reason that furrow collagens have been proposed to be part of a damage sensor that relays information about cuticle integrity to the epidermis (*Dodd et al., 2018*). Interestingly, this reporter is also induced at the late L4 stage in the wild-type before the last moult when the cuticle is reshaping (*Figure 6—figure supplement 1A*; *Miao et al., 2020*).

We examined the consequences of knocking down the expression of all these different collagen genes on the pattern of meiosomes in the adult. Collagen inactivation was always confirmed by observing the effect on body size, as well as the change in the expression of *nlp-29*p::GFP in parallel experiments (*Figure 6—figure supplement 1B and C*). Compared to control RNAi or to annuli collagen inactivation (*dpy-4*, *dpy-5* and *dpy-13*), inactivation of all six furrow collagen genes (*dpy-2*, *dpy-3*, *dpy-7*, *dpy-8*, *dpy-9*, and *dpy-10*) provoked a marked and specific alteration in the pattern of VHA-5::GFP. The meiosomes' normal reticulated pattern was fragmented, as reflected by a decrease in their average size and Feret's diameter, and a >25% increase in their density (*Figure 6B and C*). A similar fragmentation was observed with the different VHA-5 reporter strains, and either following inactivation of the furrow collagen gene's expression with RNAi or in null mutants (*Figure 6—source data 1*).

To test if the fragmentation was associated with a change in VHA-5's relation to other organelles, we inactivated furrow collagen genes in strains combining VHA-5::GFP and different mScarlet-tagged vesicular membrane markers, LGG-1 for autophagosomes, HGRS-1 for multivesicular bodies, and SNX-1 for recycling endosomes. In contrast to VHA-5::GFP, we observed no marked alteration in their patterns, and there was still no overlap with VHA-5::GFP (*Figure 6—figure supplement 2A*). Further, neither *dpy-3* nor *dpy-7* inactivation had any effect on the size and density of the vesicular pattern of EEA-1::GFP (*Figure 6—figure supplement 2B*), a marker of early endosomes (*Shi et al., 2009*). Thus, loss of furrow collagen gene leads to a substantial fragmentation of meiosomes, without affecting vesicular organelles in the epidermis. This suggests that furrow collagens play an important and specific role in maintaining meiosome integrity.

## Furrow mutants, with small meiososomes, display a detached cuticle

The cuticle is connected, through the epidermis, to the underlying body-wall muscles via hemidesmosomes. The maintenance of hemidesmosome integrity is vital; their complete loss causes a fully penetrant lethality. On the other hand, partial loss of the hemidesmosome component MUA-3 causes the cuticle to detach above the muscles (*Bercher et al., 2001*). We have shown above that hemidesmosomes and meiosomes are present in complementary and non-overlapping regions of the epidermis

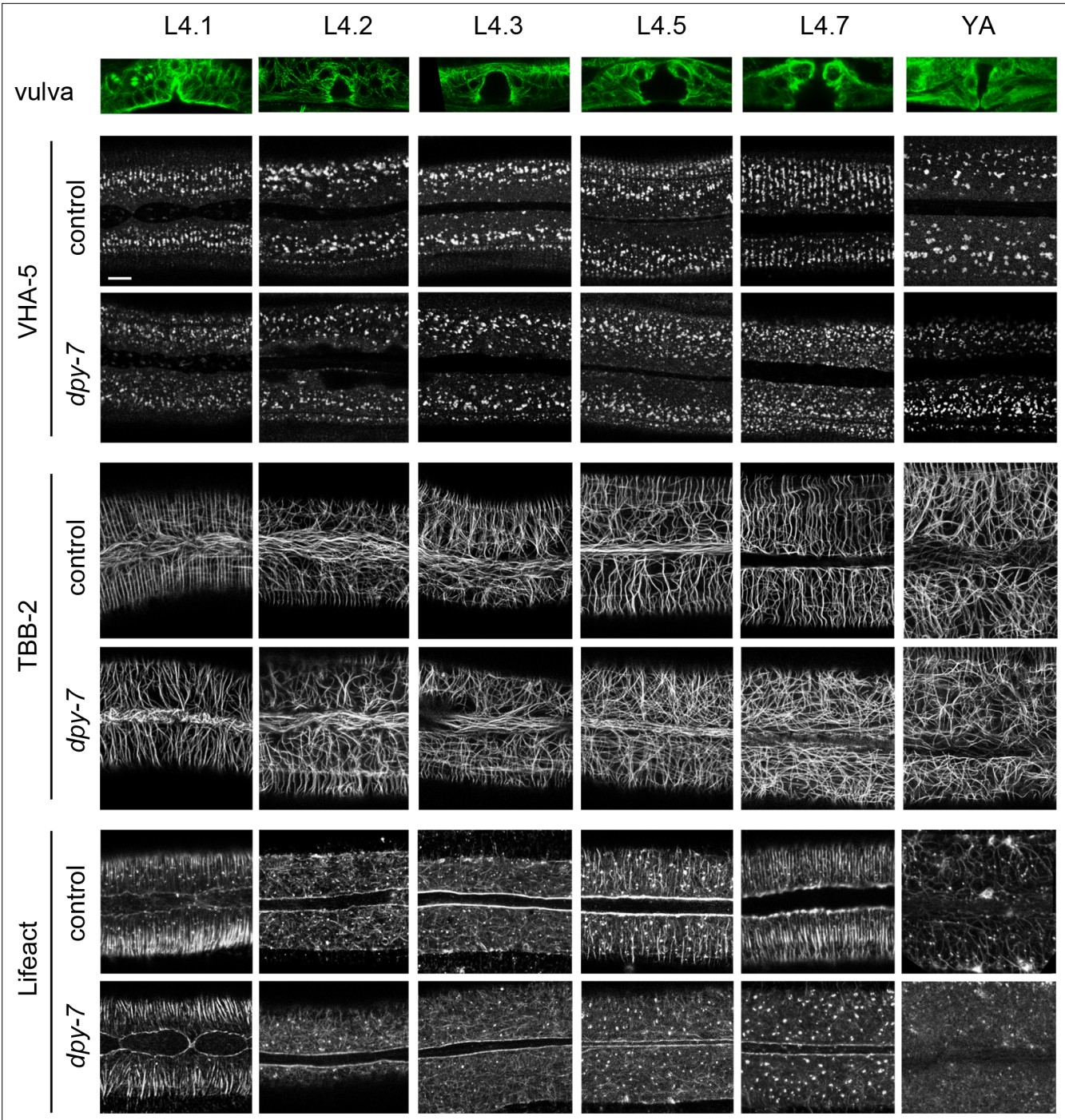

**Figure 5.** Furrow collagen inactivation provokes a loss of meisosome and cytoskeleton alignment during the L4 stage. Confocal images of worms expressing VHA-5::GFP [Ex] (upper paired panels), TBB-2::GFP (middle paired panels), and LIFEACT::GFP (lower paired panels) from early L4 to young adult (YA) stage, treated with the control (*sta-1*) or furrow Dpy (*dpy-7*) RNAi clones, n > 4. Scale bar, 5 µm. To define the precise L4 stage, the vulva was observed and worms classified according to *Cohen et al., 2020*. A representative example of the vulva at each stage is shown on the top row in worms expressing the marker TBB-2::GFP.

The online version of this article includes the following figure supplement(s) for figure 5:

**Figure supplement 1.** Furrow collagen inactivation provokes a loss of meisosomes alignment during the L4 stage.

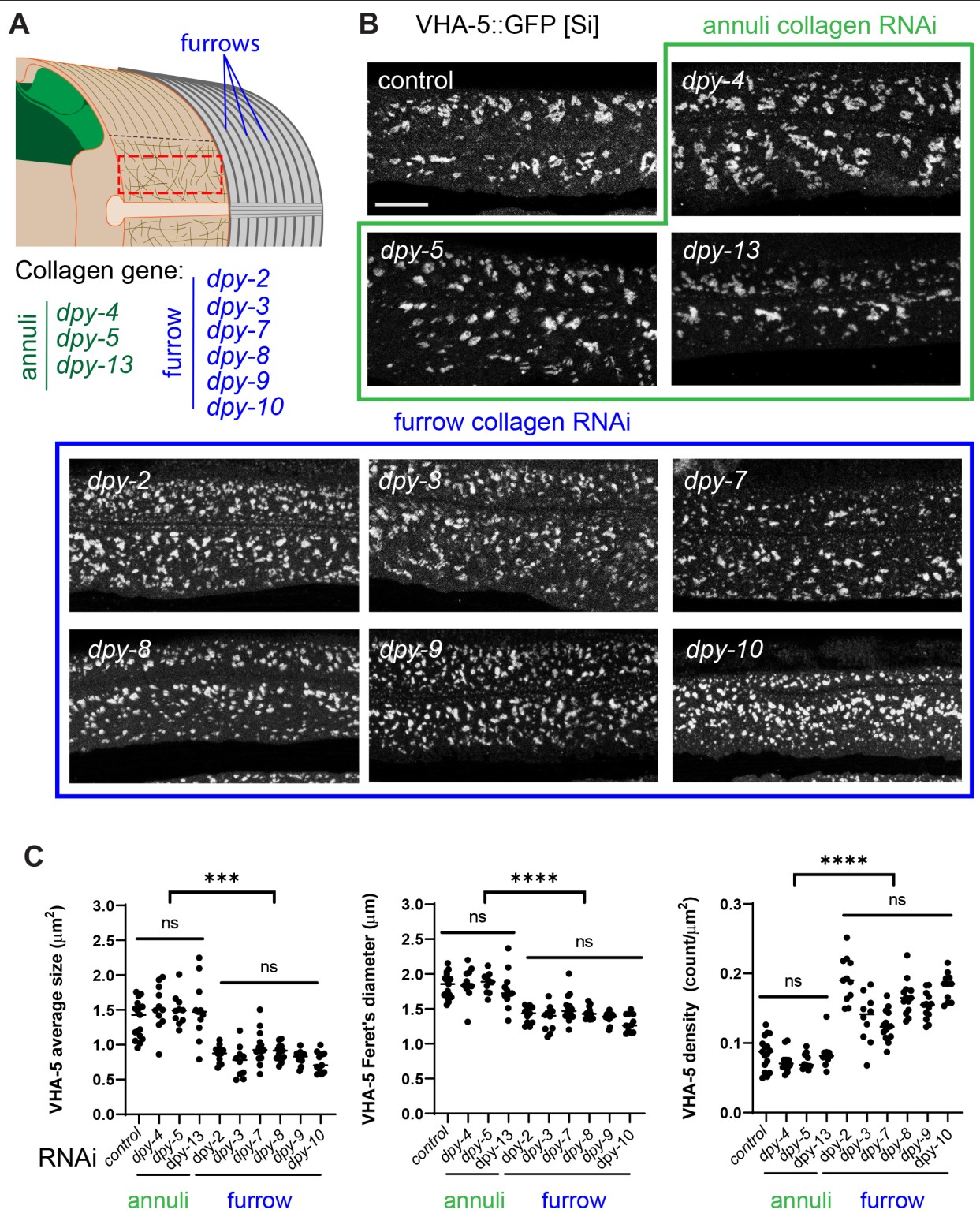

**Figure 6.** Furrow collagen inactivation provokes meiososome fragmentation. (**A**) Schematic view of the outer tissues of an adult *C. elegans* worm near the mid-body, highlighting the furrows that separate the annuli, the different collagen genes in the annuli (green) or furrow (blue) and indicating the region analysed with the red dashed rectangle. (**B, C**) L1 larvae expressing VHA-5::GFP [Si] were treated with the indicated RNAi clones and analysed at the young adult stage: control (*sta-1*), annuli collagens *dpy-4*, *dpy-5* and *dpy-13* (green) and the furrow collagens: *dpy-2, dpy-3, dpy-7, dpy-8, dpy-9,* or *dpy-10* (blue). (**B**) Confocal images of selected regions on the lateral epidermis, at a constant position relative to the vulva, scale bar, 5 µm. (**C**) Quantification of VHA-5 signal average size, Feret's diameter and density. All statistics are in *Figure 6—source data 1*, ****p<0.0001 and ***p<0.001. Control: *sta-1* (n

*Figure 6 continued on next page*

*Figure 6 continued*

= 17), *dpy-4* (n = 11), *dpy-5* (n = 10), *dpy-13* (n = 11); furrow Dpy: *dpy-2* (n = 11) *dpy-3* (n = 10), *dpy-7* (n = 14), *dpy-8* (n = 13), *dpy-9* (n = 13), or *dpy-10* (n = 11).

The online version of this article includes the following source data and figure supplement(s) for figure 6:

**Source data 1.** Quantification of the fragmentation of the meisosomes.

**Figure supplement 1.** Worms increase AMP gene expression at the late L4 stage and in furrow collagen mutants.

**Figure supplement 2.** Inactivation of furrow Dpy do not change the VHA-5 relationship to endosomes and MVB, nor affect early endosomes.

(*Figure 3B–D*). This suggests that hemidesmosomes cannot play a role in the attachment of the lateral epidermis and the dorso/ventral ridge to the cuticle. As the meisosomes, containing numerous folds of the plasma membrane, increase the surface contact between the epidermis and the cuticle, we asked whether meisosome fragmentation could impact the attachment of the epidermis to the cuticle. Examination by TEM first confirmed that the meisosomes are significantly smaller in furrow collagen mutants compared with another Dpy mutant (*dpy-13*) or the wild-type, irrespective of the direction of the section, as both longitudinal and sagittal sections show the same phenotype (*Figure 7A–D*, *Table 1*, and *Figure 1* for the wild-type). In all furrow collagen mutants examined, there was a frequent disruption of the contact between the epidermal plasma membrane and the cuticle, either in the lateral epidermis or the dorso or ventral ridge, but not above the muscle quadrants (*Figure 7B and G*, *Figure 7—figure supplement 1*). This detachment is clearly distinct from what is observed in so-called Blister mutants where, due to the absence of the connective struts, the detachment happens between the two main layers of the cuticle (*Page and Johnstone, 2007*). To confirm the phenotype, we analysed by SBF-SEM entire transversal worm sections over a length of 21.5 and 34.4 μm, for a wild-type and a *dpy-2* mutant young adult worm, respectively. This confirmed that the detachment between the cuticle and the epidermis was always found in the furrow collagen mutant outside the region of the body-wall muscles (*Figure 7H*). In furrow collagen mutants, the space between the cuticle and the underlying epidermal cell was often filled with a diverse range of cytoplasmic content, including membrane-bound vesicles with the appearance of endosomes, lysosomes, mitochondria, as well as electron-dense particles the size of ribosomes (*Figure 8A*).

To exclude the remote possibility that this detachment was an artefact linked to the different fixation protocols used for electron microscopy, we carried out live imaging on two independent strains in which the cuticle was labelled with a collagen tagged with mScarlet (ROL-6::mScarlet [KI]) and the epidermal plasma membrane was labelled with GFP::CAAX or GFP::PH-PLC1δ. Compared to the wild-type, where the GFP signal is restricted to heterogeneous macrodomains in the plasma membrane (*Taffoni et al., 2020*), in a *dpy-3* furrow collagen mutant, the GFP was seen in numerous brightly stained vesicular structures that accumulated outside the epidermis at the level of the mScarlet cuticular signal (*Figure 8B*). Together, these phenotypes lead us to hypothesise that the meisosomes may play an important role in attaching the cuticle to the underlying epidermal cell and that loss of this intimate connection causes a profound alteration of epidermal integrity.

## Furrow mutants have abnormal biomechanical properties

We predicted that the changes in cuticle attachment seen in the furrow mutants would impact the biomechanical properties of worms. It was previously shown that furrows are stiffer than the rest of the cuticle in wild-type worms (*Essmann et al., 2016*). We therefore used atomic force microscopy to measure the resistance to force in wild-type and mutant worms, as previously described (*Essmann et al., 2016*; *Essmann et al., 2020*). While topographic AFM imaging (*Figure 9A*) provided further corroboration of the fact that in the absence of furrow collagens the cuticle has a disorganised aspect with irregular folds, lacking the usual repeated linear pattern of annuli and furrows, force spectroscopy AFM revealed differences in stiffness. In contrast to the non-furrow *dpy-13* mutant that had a rigidity similar to wild-type, the different *dpy* furrow mutants (*dpy-2*, *dpy-3*, *dpy-7*, and *dpy-8*) exhibited markedly less steep force-indentation curves (*Figure 9B*), and hence lower stiffness or Young's moduli (*Figure 9C*). This suggests that furrow collagens are required for normal stiffness. While lack of certain collagens in the cuticle could directly affect cuticle stiffness, we hypothesise that the effect on stiffness is a consequence of the fact that furrow collagens are essential for the presence of normal meisosomes.

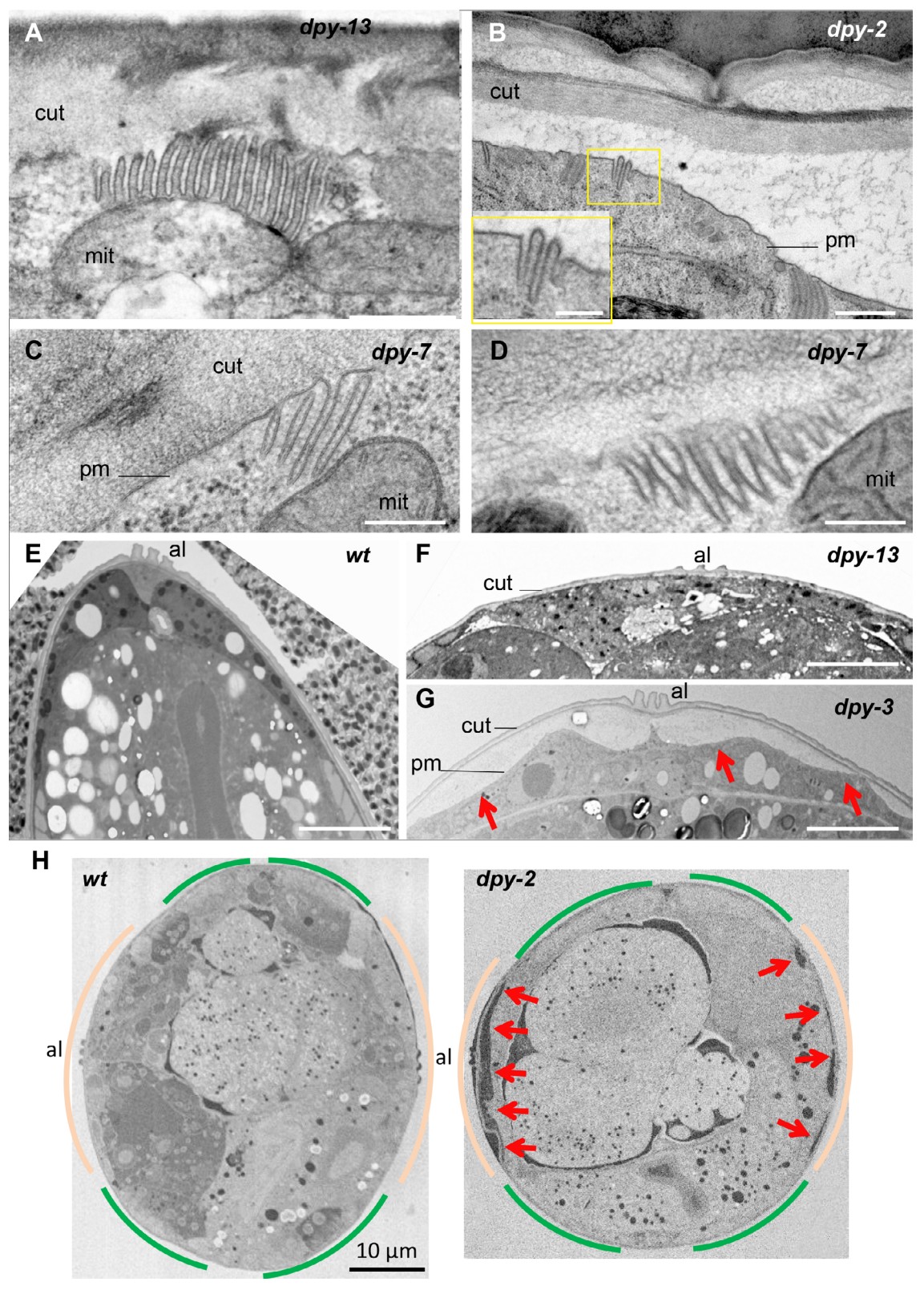

**Figure 7.** Furrow collagen inactivation leads to smaller and abnormal meiosomes and detachment of the cuticle. TEM images of young adult worms. Compared to wild-type (*Figure 1*) or *dpy-13* (**A**), *dpy-2* (**B**) *dpy-7* (**C–, D**) have abnormally small meiosomes with irregular spacing between the membrane folds (**D**); number of worms analysed are given in *Table 1*. Compared to wild-type (**E**) or *dpy-13* (**F**), lower magnification reveals detachment of the cuticle (cut) from the plasma membrane (pm) in *dpy-3* mutant worms over the whole lateral surface of the epidermal cell, on both sides of the alae

*Figure 7 continued on next page*

*Figure 7 continued*

(al) (**G**). (**H**) Compared to wild-type (left), serial block-face scanning electron microscopy (SBF-SEM) analysis of the entire transversal worm also reveals the detachment (red arrow) in a *dpy-2* mutant of the cuticle from the lateral epidermis, contrary to the regions above the muscles, delineated in beige and green, respectively (one representative slice per animal, entire transversal sections were acquired over a length of 21.5 and 34.4 µm, for a wild-type and a *dpy-2* mutant worm, respectively). Scale bar 500 nm in (**A**, **B**), 250 nm in inset in (**B**), 200 nm in (**C**, **D**), 5 µm in (**E–G**), and 10 µm in (**H**).

The online version of this article includes the following figure supplement(s) for figure 7:

**Figure supplement 1.** Furrow collagen inactivation leads to detachment of the cuticle in lateral and ventral/dorsal ridges.

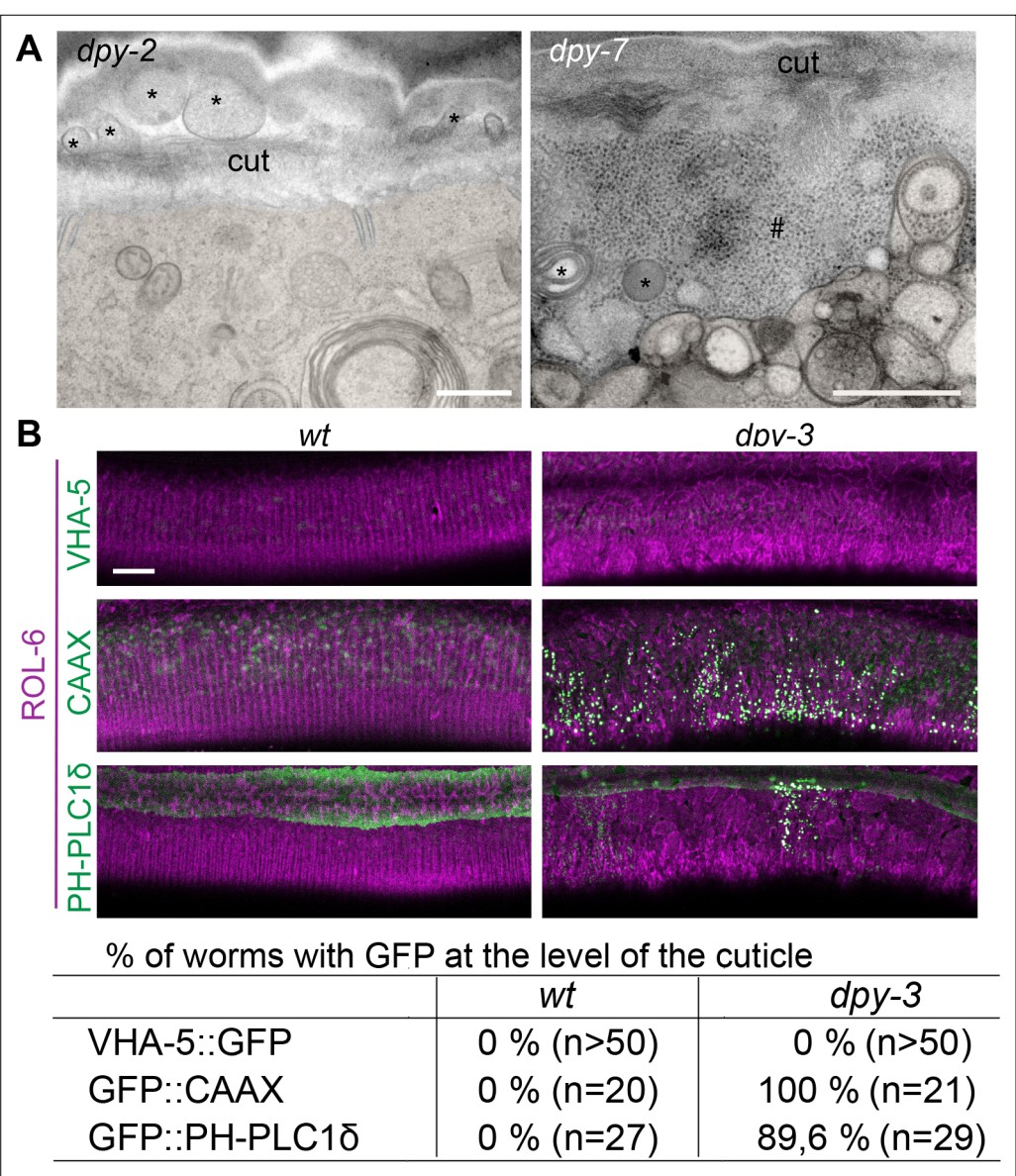

## % of worms with GFP at the level of the cuticle

| | *wt* | *dpy-3* |
|---|---|---|
| VHA-5::GFP | 0 % (n>50) | 0 % (n>50) |
| GFP::CAAX | 0 % (n=20) | 100 % (n=21) |
| GFP::PH-PLC1δ | 0 % (n=27) | 89,6 % (n=29) |

**Figure 8.** Furrow collagen inactivation provokes extrusion of membrane and cytoplasmic contents into the cuticle. (**A**) TEM images of *dpy-2* (left) and *dpy-7* (right) young adult mutant worms reveal the presence of membranous organelles (*) and cytoplasmic content, including ribosome-like particles (#), between the cuticle and the plasma membrane; epidermis is pseudo-coloured in beige. Scale bar, 500 nm. (**B**) Confocal images of wild-type (left) and *dpy-3* mutant (right) young adult worms expressing ROL-6::mScarlet and VHA-5::GFP, CAAX::GFP or PH-PLC1δ::GFP, one confocal plane was selected at the level of the cuticle using the ROL-6::mScarlet. Scale bar, 10 µm. (**C**) Quantification of the percentage of worm presenting the abnormal presence of GFP extrusion at the level of the cuticle for the different markers; number of worms observed is noted in parenthesis.

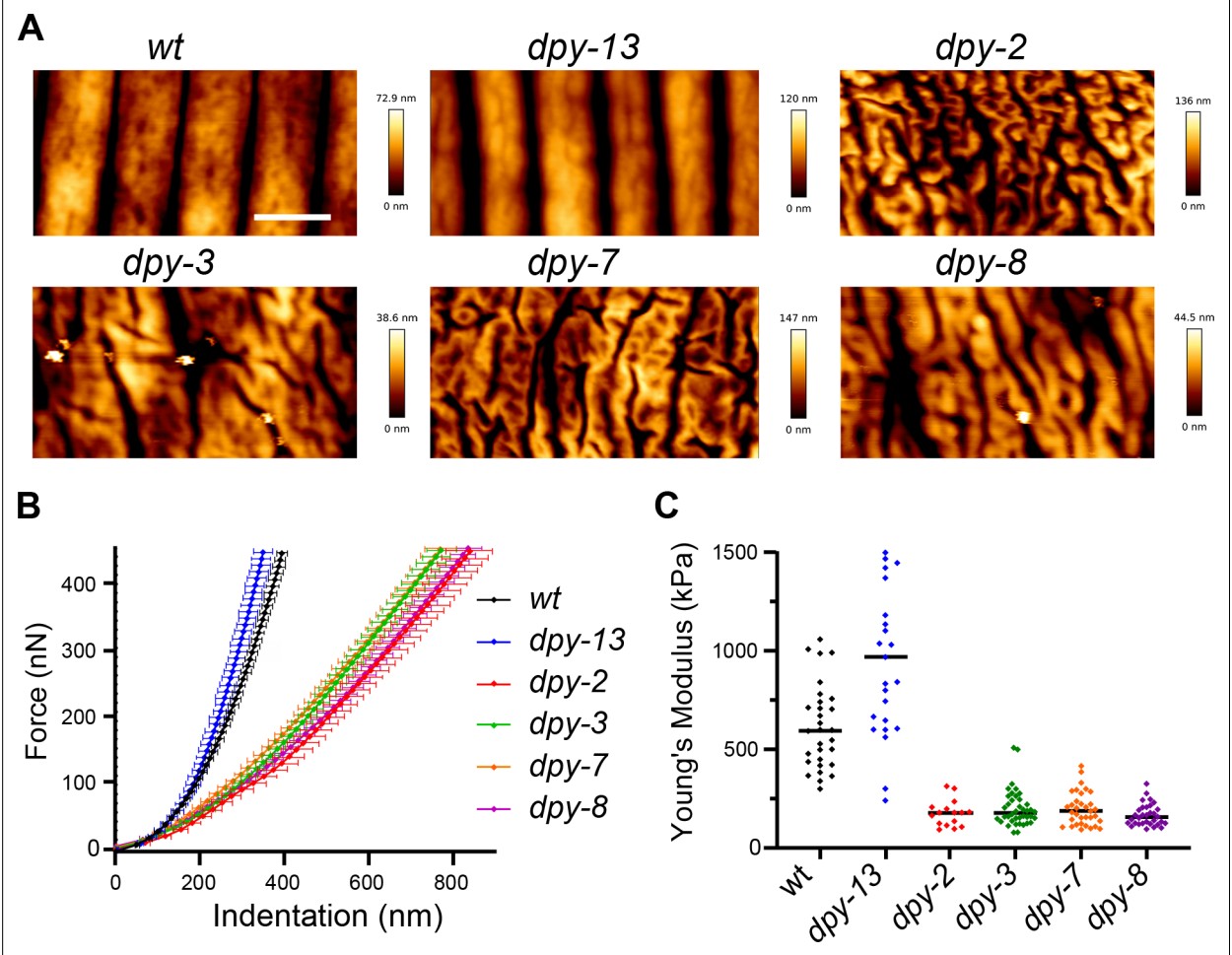

**Figure 9.** Furrow collagen inactivation provokes a reduction in stiffness of the cuticle. (**A**) AFM topography of the cuticle in wild-type, *dpy-13*, *dpy-2*, *dpy-3*, *dpy-7*, and *dpy-8* mutant adult worms. Scale bar, 1 μm. (**B**) Mean force-indentation curves of wild-type and collagen mutants acquired by AFM. (**C**) Young's modulus estimation from force curves by applying the Hertz model for contact mechanics. Data are from three independent experiments with a total of n = 30, 25, 16, 40, 34, and 32, for wild-type, *dpy-13(e184)*, *dpy-2(e8)*, *dpy-3(e27)*, *dpy-7(e88)*, and *dpy-8(e130)* mutant worms, respectively.

## Discussion

In this study, we undertook the characterisation of meisosomes, structures at the interface of the epidermis and the cuticle in *C. elegans*. Across species, interfaces exist between flexible and dynamic cell membranes and more rigid extracellular matrices. Because of requirements for growth, signal transduction, and repair, the microstructures of the ECM need to be tightly linked to the plasma membrane and cytoskeleton of the underlying cell (*Chebli et al., 2021*). In yeast, eisosomes are single-membrane invaginations located under the cell wall that bridge this boundary and fulfil this function. They can disassemble in minutes to buffer changes in membrane tension, protecting cells from osmotic shock, but also activate membrane stress signalling pathways through the release of BAR domain containing proteins (*Appadurai et al., 2020*; *Lemière et al., 2021*). Eisosome-like structures are conserved throughout fungi, microalgae, and lichens (*Lee et al., 2015*; *Zahumensky and Malinsky, 2019*), but there are no direct orthologues for core components, such as Pil1 or LSP-1, in animals. Conversely, the meisosomes that we describe here in *C. elegans*, with their multiple membrane invaginations that individually are similar in appearance to eisosomes, are, to the best of our knowledge, distinct from interfacial structures in non-nematode species. Interestingly, we show here that they are enriched in a PH-PLCδ marker, which is known to bind phosphatidylinositol 4,5-bisphosphate (PIP$_2$) (*Lemmon et al., 1995*). PIP$_2$ has a major role in signal transduction and in regulating cellular processes including actin cytoskeleton and membrane dynamics (*Katan and Cockcroft, 2020*). Moreover, we have previously shown that the same PH-PLCδ marker rapidly

reorganises upon wounding of the lateral epidermis (*Taffoni et al., 2020*). So, it is tempting to propose that analogous to eisosomes, meisosomes could have a role as a signalling platform in response to stress.

While the presence of meisosomes had been noted in earlier studies (*Hyenne et al., 2015*; *Liégeois et al., 2006*), we have been able to go beyond their previous characterisation, in part because of improvements in electron microscopy techniques. Specifically, we adapted the fixation protocol after high-pressure freezing to have a better membrane contrast in serial block scanning electron microscopy, allowing semi-automated in silico segmentation of meisosomes. Moreover, adapting a CLEM protocol, we were able to match the VHA-5::GFP observed by fluorescence microscopy to meisosomes revealed by tomography. VHA-5, together with RAL-1, are currently the only known meisosome components. In contrast to the well-defined roles of these two proteins in alae formation and exosomes biogenesis (*Hyenne et al., 2015*; *Liégeois et al., 2006*), their function in meisosomes remains to be characterised. Notably, inactivation of *ral-1* did not eliminate VHA-5::GFP fluorescence in the epidermis (*Hyenne et al., 2015*), and knocking down the expression of *vha-5* did not affect the secretion of DPY-7 (*Liégeois et al., 2006*). This suggests that the V-ATPase on meisosomes is not involved in cuticle synthesis. Further study will be required to determine the catalogue of proteins that are needed for meisosome formation and maintenance. This would then allow the function of meisosomes to be addressed directly.

Notably, a recent study reported the isolation of mutants with an abnormal pattern of VHA-5::RFP in the epidermis but attributed this to a change in MVBs (*Shi et al., 2022*), despite a lack of substantial co-localisation with HGRS-1, a well-characterised MVBs marker, part of the ESCRT-0 complex that sorts endosomes to MVBs (*Babst, 2011*). Since previous studies (*Liégeois et al., 2006*), and the results presented here, show that VHA-5 is predominantly a marker of meisosomes, more so than of MVBs, these mutants with an abnormal pattern of VHA 5::RFP more likely affect meisosomes. Indeed, we hypothesise that the one gene that was characterised in detail (*Shi et al., 2022*), *fln-2*, which encodes the F-actin cross-linking protein filamin (*Zhao et al., 2019*), could actually be involved in the formation and/or maintenance of meisosomes. Interestingly, a *fln-2* loss-of-function mutation has been serendipitously found in several *C. elegans* strains originating from a different wild-type stocks (*Zhao et al., 2019*), so careful attention to genotypes will be needed in future work. Regardless, *fln-2* may represent an important tool to investigate meisosome function.

Setting this issue aside, by taking an ultrastructural approach, we were able to build up a detailed picture of the organisation of meisosomes. One of their defining features is the constant 35 nm spacing of their constituent plasma membrane folds. This raises the question of how the membrane folds with such precision. One possibility is that the striking electron-dense material that is apposed to each side of the membrane on the cytoplasm-facing folds, spaced less than 10 nm apart, will contain specific structural protein that maintain the uniform width of each meisosome fold and influence their mechanical properties. These structures will require more precise characterisation. We equally have yet to establish whether the frequent proximity of meisosomes to mitochondria, with a close apposition of membranes, has a functional significance.

Contrary to the cuticle of many adult insects, the nematode cuticle is flexible enough to allow bending during locomotion. It is also thought to stretch to accommodate growth between moults. When the old cuticle is shed, it leaves in its place the new cuticle that had been moulded underneath it. The circumferential furrows of the new cuticle thus appear exactly in register with the position of old furrows. Before moulting, the cytoskeleton aligns in the apical epidermis, underneath, and parallel to each furrow. Although this had been proposed to be important for positioning the furrows of the new cuticle (*Costa et al., 1997*; *McMahon et al., 2003*; *Page and Johnstone, 2007*), a recent study found unexpectedly that actin is dispensable for the alignment of furrows (*Katz et al., 2018*). On the other hand, we found that the furrows are required for the alignment of actin fibres before the last moult. We propose therefore that only the old furrows are required to pattern the new furrows. Consistent with such a model, the LPR-3 protein that is part of the transient pre-cuticle that is formed between the old and the new cuticles before each moult is absent from the region of the furrows (*Forman-Rubinsky et al., 2017*). We have shown that furrow determines the regular parallel and circumferential positioning of meisosomes. It is not yet clear whether this alignment of meisosomes is functionally important. It could result from steric constraints during moulting, in the limited space between nascent furrows of the new cuticle and the closely apposed circumferential actin fibres. It

should, however, be noted that this alignment is not seen for vesicular organelles like MVBs, endosome, or autophagosomes.

As adults, furrowless collagen mutants have fragmented meisosomes and a detached cuticle. Although this fragmentation could be a consequence of the detachment, we favour the converse hypothesis that fragmentation causes the detachment, and that the multiple folds of plasma membrane normally increase its contact surface with the cuticle thus ensuring a more robust connection of the aECM to the lateral epidermis. While the lateral epidermis is rich in meisosomes, it is devoid of hemidesmosomes. Conversely, in the dorsal and ventral quadrants, there are essentially no meisosomes, but abundant hemidesmosomes. These latter structures secure the muscles to the cuticle through epidermis and are indispensable for worm development and viability. Above the muscles, the epidermis is extremely thin, with the apical and basal plasma membranes juxtaposed, linked via intermediate filaments that bridge apical and basal hemidesmosome protein complexes (*Zhang and Labouesse, 2010*). MUA-3 is a hemidesmosome transmembrane protein in direct contact with the cuticle. In hypomorphic *mua-3* mutants, large gaps form between the apical epidermal surface and the cuticle in the dorsoventral quadrants, reflecting a loss of attachment of apical hemidesmosomes to the cuticle. Unlike the cytoplasm-filled gaps we observed in furrowless mutants, in *mua-3(rh195)* worms, these spaces appear devoid of contents, and the apical epidermal membrane is intact (*Bercher et al., 2001*). So, in contrast to the loss of hemidesmosomes, fragmentation of meisosomes in furrowless mutants affects the integrity of the apical epidermal membrane in the lateral epidermis, potentially explaining the permeability phenotype of furrowless mutants (*Sandhu et al., 2021*). Despite these differences, both meisosomes in the lateral epidermis, and hemidesmosomes in the dorsoventral quadrants, do appear to have an analogous function, ensuring the attachment of the apical plasma membrane to the cuticle.

In animals, ECMs provide mechanical support for tissue assembly and organ shape. During embryogenesis in *C. elegans*, the aECM is essential during elongation as it not only maintains embryonic integrity, but also relays the mechanical stress produced by the actomyosin cytoskeleton and the muscles (*Vuong-Brender et al., 2017a*; *Vuong-Brender et al., 2017b*). In the adult, the mechanical properties of the aECM have only recently started to be explored. Atomic force microscopy revealed that the furrows have a higher stiffness than the annuli (*Essmann et al., 2016*). Here, we show that loss of specific furrow collagens, but not of non-furrow collagens, decreases stiffness. Part or all of this could be a direct consequence of the altered cuticle morphology, an analogy being the increased stiffness that corrugation provides. Furrow Dpy mutants are known to have a higher internal concentration of glycerol (*Wheeler and Thomas, 2006*), which will decrease their internal hydrostatic pressure. We propose that this decreased hydrostatic pressure is a consequence of the decrease in the stiffness of the cuticle. It would ensure the necessary balance of inward and outward pressures required for body integrity. Since we used a 10 µm diameter AFM probe to indent the worm, and the indentation depth was greater than the thickness of the cuticle (ca. 800 nm compared to 500 nm for the cuticle), our measurements did not directly assess the cuticle stiffness, so further investigations will be needed to confirm our hypothesis. It is interesting to note, however, that a decrease in stiffness and an increase in the activity of innate immune signalling pathways in the epidermis are signatures of ageing in *C. elegans* (*Lezi et al., 2018*; *Essmann et al., 2020*). How physiological and pathological modifications of the biomechanical properties of the aECM are surveyed by the epidermis remains an open question for future studies.

## Materials and methods
### Nematode strains
All *C. elegans* strains were maintained on nematode growth medium (NGM) and fed with *E. coli* OP50, as described (*Stiernagle, 2006*), the wild-type N2, IG274 *frIs7[col-12p::DsRed, nlp-29p::GFP] IV* (*Pujol et al., 2008a*), IG1697 *dpy-2(e8) II; frIs7[nlp-29p::GFP, col-12p::DsRed] IV*, IG1685 *dpy-3(e27) X; frIs7[nlp-29p::GFP, col-12p::DsRed] IV*, IG1689 *dpy-7(e88) X; frIs7[nlp-29p::GFP, col-12p::DsRed] IV* (*Dodd et al., 2018*), IG1699 *dpy-8(e130) X; frIs7[nlp-29p::GFP, col-12p::DsRed] IV*, IG344 *dpy-13(e184) frIs7[nlp-29p::GFP, col-12p::DsRed] IV*, RT424 *pwIs126[eea-1p::GFP::EEA-1* (*Shi et al., 2009*), RT3657 *pwSi46[hyp7p::mScarlet::HGRS-1+G418R]*, RT3640 *pwSi65[hyp7p::mScarlet::SNX-1+G481R]*, RT3635 *pwSi62[hyp7p::mScarlet::LGG-1+G418R]* (*Serrano-Saiz et al., 2020*), ML2113

*mcIs67[dpy-7p::LifeAct::GFP; unc-119(+)] V; stIs10088[hlh-1::his-24::mCherry, unc-119(+)]* (*Lardennois et al., 2019*) to visualise actin in larval stages, IG1813 *frSi9[pNP151(col-62p::Lifeact::mKate_3'c-nmy), unc-119(+) ttTi5605]II; tbb-2(tj26[GFP::TBB-2]) III* (*Taffoni et al., 2020*) to visualise actin in the adult, IG1935 *frSi9[pNP151(col-62p::Lifeact::mKate_3'c-nmy), unc-119(+) ttTi5605] II; Is[wrt-2p::GFP::PH-PLC1δ, wrt-2p::GFP::H2B, lin-48p::mCherry]*, XW18042 *qxSi722[dpy-7p::DPY-7::sfGFP; ttTi5605] II* (*Miao et al., 2020*) and MBA365 *Ex[dpy-7p::GFP::CAAX, myo-2p::GFP]* kindly provided by M. Barkoulas (UCL).

Extrachromosomal transgenic strain [Ex] containing GFP tagged version of VHA-5 were generated by microinjection in N2 worms of the *vha-5p*VHA-5::GFP construct pML670 (*Liégeois et al., 2006*) kindly provided by M. Labouesse (LBD/IBPS) at 3 ng/µl together with *unc-122*p::GFP at 100 ng/µl to generate IG1930. The transgene *frSi26* is a single-copy insertion on chromosome II (ttTi5605 location) of pNP165 (*dpy-7*p::VHA-5::GFP) by CRISPR using a self-excising cassette (SEC) (*Dickinson et al., 2015*). pNP165 was obtained by insertion of the *dpy-7* promoter, which leads to an epidermal specific expression, in front of VHA-5::GFP into the pNP154 vector. pNP154 was made from a vector containing the SEC cassette for single insertion on chromosome II at the position of ttTi5605 (pAP087, kindly provided by Ari Pani) (*Watts et al., 2020*). Constructs were designed using the plasmid editor Ape (*Davis and Jorgensen, 2022*) and made using Gibson Assembly (NEB Inc, MA) and confirmed by sequencing. pNP165 was injected in N2 at 20 ng/µl together with pDD122 (*eft-3*p::Cas9) at 50 ng/µl, pCFJ90 (*myo-2*p::mCherry) at 2 ng/µl, and #46168 (*eef-1A.1*p::CAS9-SV40_NLS::3'*tbb-2*) at 30 ng/ml. pCFJ90 was a gift from Erik Jorgensen (Addgene plasmid # 19327; http://n2t.net/addgene:19327; RRID:Addgene_19327) (*Frøkjaer-Jensen et al., 2008*). Non-fluorescent roller worms were selected then heat shocked to remove the SEC by FloxP as described in *Dickinson et al., 2015* to generate IG2118 *frSi26[pNP165(dpy-7p::VHA-5::GFP) ttTi5605] II*. Fluorescent knock-in [KI] reporter strains were generated through CRISPR editing (SunyBiotech) at the C-terminus of the gene to generate PHX5715 *vha-5(syb5715[VHA-5::sfGFP]) IV*, which was further outcrossed two times to generate IG2144, and PHX2235 *rol-6(syb2235[ROL-6::mScarlet]) II*. All the multiple reporter strains generated in this study were obtained by conventional crosses (see *Supplementary file 1* for a list of all strains).

## Transmission electron microscopy (TEM)

Day 1 adult worms were frozen in NaCl 50 mM medium containing 5% of BSA and *Escherichia coli* bacteria using Leica EM Pact 2 high-pressure freezer. After freezing, samples were freeze-substituted at –90°C in acetone containing 2% $OsO_4$ for 96 hr. The temperature was gradually increased to –60°C and maintained for 8 hr. The temperature was then raised to –30°C and maintained for 8 hr, before to be raised again to room temperature (RT). Samples were finally washed in acetone and embedded in epoxy resin. Resin was polymerised at 60°C for 48 hr. Then, 70 nm ultrathin and 350 nm semithin sections were performed using a Leica UC7 ultramicrotome and post-stained with 2% uranyl acetate and Reynolds' lead citrate. Images were taken with a Tecnai G2 microscope (FEI) at 200 kV. For tomography acquisitions, tilted images (+60°/–60° according to a Saxton scheme) were acquired using Xplorer 3D (FEI) with a Veleta camera (Olympus, Japan). Tilted series alignment and tomography reconstruction was performed using IMOD (*Mastronarde and Held, 2017*).

## Freeze fracture

Wild-type adults were fixed in buffered 2.5% glutaraldehyde, then cryoprotected in 30% glycerol overnight prior to freezing. Fixed animals were positioned between two gold discs, and plunge frozen in liquid nitrogen-chilled isopentane. Frozen worms were placed into a double replica holder for a Balzer's 301 freeze etch device. Samples were cleaved within the freeze etch device by free breaks, then shadowed with Pt/C to form a metal replica. Replicas were washed in bleach to remove all tissue prior to mounting on slot grids for examination by TEM.

## Scanning electron microscopy by serial block face (SBF-SEM)

After freezing in the aforementioned conditions, samples were incubated at –90°C in acetone containing 2% $OsO_4$ for 106 hr. The temperature was gradually increased to 0°C and samples were washed over 1 hr in acetone at RT. Samples were then incubated in acetone containing 0.1% TCH for 60 min, washed over 1 hr in acetone, and incubated in acetone containing 2% $OsO_4$ for 1 hr at RT. After rehydration in ethanol decreasing graded series, samples were incubated ON in 1% aqueous

uranyl acetate at 4°C and in 30 nM lead aspartate for 30 min at 60°C. Samples were finally dehydrated in graded series of ethanol baths and pure acetone and embedded in Durcupan resin. Resin was polymerised at 60°C for 48 hr. For the segmentation of meisosomes, regions of the lateral epidermis were acquired over a length of 12 μm, with a resolution of 10 nm par pixels. For scanning the cuticle detachment, entire transversal sections were acquired over a length of 21.5 and 34.4 μm, for a wild-type and a *dpy-2* mutant worm, respectively, with a resolution of 10 nm par pixels.

### Correlative light electron microscopy (CLEM)

Sample for CLEM experiments were treated as in *Johnson et al., 2015*. Briefly, the worms were high-pressure frozen (EMPACT2, Leica) and then freeze-substituted (AFS2, Leica) for 20 hr from –130°C to –45°C in an acetone-based cocktail containing 0.2% uranyl acetate, 0.1% tannic acid, and 5% $H_2O$. After 2 hr of acetone washes at –45°C, the samples were infiltrated with gradients of HM20 resin over 9 hr, with pure resin for 18 hr at –45°C and the resin was polymerised under UV for 24 hr at –45°C and for 12 hr at 0°C. Then 350 nm semithin sections were processed as described for TEM tomography above. TEM grids were first analysed at by confocal imaging where a bright-field image is overlaid with the fluorescent image, then analysed in TEM at low magnification. Brightfield, GFP confocal, and TEM images were aligned using Amira software. Several positions with two or three GFP spots were chosen to do a high-magnification tomography, as described above, to reveal the meisosomes.

### Segmentations and 3D image analysis

For electron tomography datasets, a binned version of the reconstructed tomogram was segmented using the Weka 3D segmentation plugin in Fiji/ImageJ to visualise the mitochondria and the meisosomes. The cuticle was visualised by the Amira-embedded Volume Rendering plugin from a manually segmented mask. A cropped area of interest of the full resolution electron tomogram was segmented in iLastik to visualise a representative portion of the organelle. For serial block face datasets, the segmentation of the meisosome and mitochondria was generated using the Weka 3D segmentation plugin in Fiji/ImageJ. Animations and snapshots were generated in Amira.

### RNA interference

RNAi bacterial clones were obtained from the Ahringer or Vidal libraries and verified by sequencing (*Kamath et al., 2003*; *Rual et al., 2004*). RNAi bacteria were seeded on NGM plates supplemented with 100 μg/ml ampicillin and 1 mM isopropyl-β-D-thiogalactopyranoside (IPTG). Worms were transferred onto RNAi plates as L1 larvae and cultured at 20°C or 25°C until L4 or young adult stage. In all our experiments, we are using *sta-1* as our control, as we have shown over the last decade that it does not affect the development nor any stress or innate response in the epidermis (*Dierking et al., 2011*; *Lee et al., 2018*; *Taffoni et al., 2020*; *Zhang et al., 2021*; *Zugasti et al., 2014*; *Zugasti et al., 2016*).

### Fluorescent image acquisition

Live young adult worms were placed on a 2% agarose pad containing 0.25 mM levamisole in NaCl to immobilise the worms. Images were acquired using a confocal laser scanning microscopy: Zeiss LSM780 and its acquisition software Zen with a Plan-Apochromat ×40/1.4 Oil DIC M27 objective with a zoom 2–4, a Plan-Apochromat ×63/1.40 Oil DIC M27 with a zoom 1. Spectral imaging combined with linear unmixing was used in most confocal images to separate the autofluorescence of the cuticle.

### Airyscan super-resolution microscopy

Airyscan imaging was performed using a commercial Zeiss confocal microscope LSM 880 equipped with an Airyscan module (Carl Zeiss AG, Jena, Germany) and images were taken with a ×63/1.40NA M27 Plan Apochromat oil objective. In this mode, emission light was projected onto an array of 32 sensitive GaAsP detectors, arranged in a compound eye fashion. The Airyscan processing was done with Zen Black 2.3 software by performing filtering, deconvolution, and pixel reassignment to improve SNR. The Airyscan filtering (Wiener filter associated with deconvolution) was set to the default filter setting of 6.1 in 2D.

### Fluorescent image analysis

To extract the morphological properties of meisosomes, we devised an automatic Fiji segmentation procedure (GitHub; https://github.com/centuri-engineering/BD_BlobsSeg, copy archived at

*Dehapiot, 2022*). We first restricted the analysis to manually drawn ROIs and isolated organelles (foreground image) from the background by using the 'remove outliers' function of Fiji (radius = 30 pixels and threshold = 30). We next applied a Gaussian blur (sigma = 1 pixel) on the foreground image and automatically defined a threshold value to binarize the newly blurred image. This threshold was determined automatically by multiplying the background value (retrieved by averaging the fluorescent levels of the background image) by a constant coefficient. This allowed us to normalise the segmentation since the expression levels of fluorescent proteins may vary from one animal to another. Finally, after filtering out smaller objects (less than ~0.15 µm²), we measured the averaged organelles area, Feret's diameter (longest axis), and density in the different conditions. Unpaired t-test was used to compare the samples that passed the normality test (Shapiro–Wilk normality test) and with homogeneity variances (Fisher test) and unpaired nonparametric Mann–Whitney test for the others. For co-localisation analysis, we counted the percentage of segmented objects in a given channel, GFP (G) or RFP (R), whose centroid is located in an object of the other channel. We then averaged these percentages across images, each representing a different worm (n = 10 for each strain analysed).

## Fluorescent reporter analyses

Analysis of *nlp-29*p::GFP expression was quantified with the COPAS Biosort (Union Biometrica; Holliston, MA) as described in *Labed et al., 2012*. In each case, the results are representative of at least three independent experiments with more than 70 worms analysed. The ratio between GFP intensity and size (time of flight [TOF]) is represented in arbitrary units. Fluorescent images were taken of transgenic worms mounted on a 2% agarose pad on a glass slide anaesthetised with 0.01% levamisole using the Zeiss AxioCam HR digital colour camera and AxioVision Rel. 4.6 software (Carl Zeiss AG).

## Atomic force microscopy (AFM)

Worms were prepared as described before (*Essmann et al., 2016*). Briefly, staged 1-day-old young adult worms were paralysed in 15 mg/ml 2, 3-butanedione monoxime (Sigma) for 2 hr at RT, and transferred to an ~2-mm-thick 4% agarose bed in a Petri dish (30 mm). Heads and tails were fixed with tissue glue (Dermabond, Ethicon) and the dish filled with a 2.5 ml M9 buffer. AFM data of worms were obtained using a NanoWizard3 (JPK) under aqueous conditions. Type qp-CONT-10 (0.1 N/m; nanosensors) cantilevers were used for imaging in contact-mode at setpoint 0.3 V and 0.5 Hz scanning speed, and NSC12 tipless cantilevers (7.5 N/m; MikroMash) with a 10 µm borosilicate bead attached (produced by sQUBE; https://www.sqube.de/) were used in force spectroscopy mode to obtain force-indentation measures at 450 nN force setpoint and 0.5 µm/s indentation speed. Cantilever sensitivity and stiffness (k) were calibrated using the JPK system calibration tool before each experiment. AFM raw data were analysed using the JPK analysis software. All force curves were processed to zero the baseline to determine the tip-sample contact point and to subtract cantilever bending. The Young's modulus was calculated within the software by fitting the Hertz/Sneddon model respecting the indenter shape (10 µm bead) to each curve. All topographical images are flattened using the plane fitting option of the JPK software at 1–2° to correct for sample tilt and natural curvature of the worm.

## Acknowledgements

We thank Jonathan Ewbank for support and input throughout the project, Ken CQ Nguyen for some of the TEM imbedding, Michel Labouesse, Barth Grant, Michalis Barkoulas, and Ari Pani for sharing strains and reagents, Chris Crocker at Wormatlas for diagrams, Roxane Fabre for the Airyscan images, Meera Sundaram, Thomas Sontag and Michel Labouesse for critical reading of the MS, and Barth Grant and Erik Jorgensen for discussions. Worm sorting was performed by Jerome Belougne using the facilities of the French National Functional Genomics platform, supported by the GIS IBiSA and Labex INFORM. Electron tomography in Figure 1—figure supplement 3 were performed at the New York Structural Biology Center, with help from KD Derr and William Rice. We thank John White and Jonathan Hodgkin for sharing the MRC/LMB archive of nematode micrographs. Some *C. elegans* strains were provided by the CGC, which is funded by NIH Office of Research Infrastructure Programs (P40 OD010440). We acknowledge the PICsL-FBI photonic microscopy facility of the CIML (ImagImm) and the PICsL-FBI electron microscopy facility of the IBDM, members of the national infrastructure France-BioImaging supported by the French National Research Agency (ANR-10-INBS-04). The

project leading to this publication has received funding from France 2030, the French Government program managed by the French National Research Agency (ANR-16-CONV-0001) and from Excellence Initiative of Aix-Marseille University - A*MIDEX.

Work is funded by the French National Research Agency ANR-22-CE13-0037-01, ANR-16-CE15-0001-01, and ANR-10-INBS-04–01 (France Bio Imaging), by the 'Investissements d'Avenir' French Government program (ANR-16-CONV-0001) and from Excellence Initiative of Aix-Marseille University - A*MIDEX and institutional grants from CNRS, Aix Marseille University, National Institute of Health and Medical Research (Inserm) to the CIML; and by a NIH OD 010943 to DHH.

## Additional information

### Funding

| Funder | Grant reference number | Author |
|---|---|---|
| Agence Nationale de la Recherche | ANR-22-CE13-0037-01 | Nathalie Pujol |
| Agence Nationale de la Recherche | ANR-16-CE15-0001-01 | Nathalie Pujol |
| Agence Nationale de la Recherche | ANR-10-INBS-04-01 | Nicolas Brouilly Nathalie Pujol |
| Agence Nationale de la Recherche | ANR-16-CONV-0001 | Benoit Dehapiot Nathalie Pujol |
| National Institutes of Health | NIH R24OD010943 | David H Hall |

The funders had no role in study design, data collection and interpretation, or the decision to submit the work for publication.

### Author contributions

Dina Aggad, Clara Luise Essmann, Formal analysis, Investigation, Visualization; Nicolas Brouilly, Shizue Omi, Formal analysis, Investigation, Visualization, Methodology; Benoit Dehapiot, Software, Methodology; Cathy Savage-Dunn, Resources; Fabrice Richard, Chantal Cazevieille, Kristin A Politi, Investigation, Methodology; David H Hall, Funding acquisition, Investigation, Methodology; Remy Pujol, Data curation, Formal analysis, Investigation, Visualization; Nathalie Pujol, Conceptualization, Formal analysis, Supervision, Funding acquisition, Investigation, Writing - original draft, Project administration, Writing - review and editing

### Author ORCIDs

Dina Aggad http://orcid.org/0000-0002-4212-4197
Nicolas Brouilly http://orcid.org/0000-0002-4584-0164
Shizue Omi http://orcid.org/0000-0002-2711-2016
Clara Luise Essmann http://orcid.org/0000-0001-5468-9489
Benoit Dehapiot http://orcid.org/0000-0002-7559-5497
Cathy Savage-Dunn http://orcid.org/0000-0002-3457-0509
Fabrice Richard http://orcid.org/0009-0007-2748-8797
Kristin A Politi http://orcid.org/0000-0002-8974-6727
David H Hall http://orcid.org/0000-0001-8459-9820
Remy Pujol http://orcid.org/0000-0002-7837-7129
Nathalie Pujol http://orcid.org/0000-0001-8889-3197

### Decision letter and Author response

Decision letter https://doi.org/10.7554/eLife.75906.sa1
Author response https://doi.org/10.7554/eLife.75906.sa2

## Additional files

### Supplementary files
• Supplementary file 1. *C. elegans* strains used in this study.

• Transparent reporting form

### Data availability
No large datasets were generated in this study. All quantifications generated in this study are included in the manuscript and supporting files. Source Data files have been provided for Table 1 & Figures 6. The automatic Fiji segmentation procedure is on GitHub https://github.com/centuri-engineering/BD_BlobsSeg (copy archived at swh:1:rev:ada8b28bb4c5b0452eadea4ee26063205cb52bcf; *Dehapiot, 2022*).

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
