## [Editor Report]

This valuable work addresses the cellular mechanisms that mediate attachment of the lateral epidermis to the cuticle. The evidence supporting the role of structures called 'meisosomes' by the authors is solid, and addresses the roles of these structures in maintaining and patterning the epidermis and cuticle. The work will be of interest to developmental and cell biologists.

---

## [Decision Letter]

**Decision letter after peer review:**

[Editors’ note: the authors submitted for reconsideration following the decision after peer review. What follows is the decision letter after the first round of review.]

Thank you for submitting the paper "Meisosomes, folded membrane platforms, link the epidermis to the cuticle in *C. elegans*" for consideration by *eLife*. Your article has been reviewed by 3 peer reviewers, and the evaluation has been overseen by a Reviewing Editor and a Senior Editor. The following individuals involved in review of your submission have agreed to reveal their identity: David S Fay (Reviewer #3).

Comments to the Authors:

We are sorry to say that, after consultation with the reviewers, we have decided that this work will not be considered further for publication by *eLife*.

Specifically, the reviewers have raised concerns on whether VHA-5::GFP can serve as a "bona fide" meisosome marker and whether meisosomes serve as attachment platforms between the cuticle and the epidermis. A substantial amount of work needs be conducted to address the concerns raised by the reviewers, and it is unlikely that this can be done within the two-month limit set by *eLife*. If you found that the critical concerns raised by the reviewers can be addressed, we encourage you to submit a revised version that will be re-evalulated.

*Reviewer #1 (Recommendations for the authors):*

1. In order to properly study the functions of meisosomes, research models should be generated to directly and specifically disrupt meisosome structures, rather than indirectly affect meisosomes by disrupting other epidermal-related structures. Theoretically, knockdown or loss-of-function mutations of meisosome-specific structural components should serve the purpose.

2. Transgenes carrying fluorescent fusion reporters, especially multi-copy extrachromosomal arrays (such as the frEx624[pML670(VHA-5::GFP, unc-122p::GFP)] used in most figures of this manuscript), are known to misbehave and often do not faithfully reflect the endogenous distribution patterns of the protein-of-interest. Therefore, at least one alternative approach for meisosome morphological analysis should be provided to confirm the key findings of this paper. There are indeed a few TEM images of the furrow mutants showing one or two smaller meisosome structures (Figure 8), which is very good itself. Unfortunately there are no parallel-performed controls in this dataset and no quantification analysis.

3. In most RNAi experiments, knockdown of the STAT family transcription factor sta-1 was used as negative control instead of the commonly used empty RNAi clone vector L4440. However, no explanation was provided as for why this particular negative control was chosen. In fact, the gene name sta-1 did not appear at all in the entire maintext. The authors need to explain why they assume that sta-1 deficiency does not affect the epidermis and the cuticle (even indirectly, because sta-1 is expressed in the body-wall muscles adjacent to the epidermis, and defects in the muscles can greatly affect epidermal biogenesis and function), and what are the advantages of choosing sta-1 RNAi as the negative control over the empty RNAi vector or RNAi clones targeting other genes.

4. Some observations described in this paper have been previously reported by other groups, but the original papers were not cited. For example, up-regulation of nlp-29 near the molting period was first reported in Miao et al., Dev Cell, 2020. This reference should be cited alongside Figure 4—figure supplement 1A.

*Reviewer #2 (Recommendations for the authors):*

1. Apical membrane stack is well known and much better to describe this folded structure, not necessary to use "Meisosomes" to rename it. And this structure should not be defined as an organelle.

2. VHA-5 is well known to localize to the apical membrane stacks (Liegeois 2006) and could be served as marker of apical membrane structure. How does VHA-5::GFP expression, puncta size, and intensity correlate with the structure of the apical membrane stack? The information about the VHA-5::GFP transgene frEx624[pML670(VHA-5::GFP, unc-122p::GFP)] is not very clear. Is VHA-5::GFP expression under its own promoter or another promoter? It is known that the extrachromosomal array of transgene show various expression level, thus the VHA-5::GFP knock-in strain should be used to quantify the localization, puncta size, and intensity. With the CRISPR-Cas9 genome editing method, it should be easy to make knock-in strain now.

3. The authors suggest furrow collagen inactivation causes VHA-5::GFP fragmentation. First, there should be an empty vector RNAi served as a negative control. Second, all the RNAi methods should be validated to indicate the effectiveness of the knockdown or mutant animals that should be used. Third, the stable knock-in of VHA-5::GFP strain should be used for quantification.

4. The authors conclude that "They could also be involved in relaying tensile information from the cuticle to the underlying epidermis as part of an integrated stress response to injury and infection." However, the evidence is weak. There is no assay to show that the apical membrane stack can respond to injury or infection and that is involved in mechanical tension. It is also unclear how vha-5 mutation affects innate immune response or wound response.

5. What's the function of apical membrane stack in collagen secretion?

*Reviewer #3 (Recommendations for the authors):*

Comments are in (approximate) relative order:

– The term meisosomes. The authors have (re)named the folded membrane structures "meisosomes". These were previously referred to as "apical membrane stacks" by Labouesse and colleagues, although the term "meisosomes" is perhaps more efficient and descriptive. Another term that may be relevant is "Ward bodies". According to PMID: 23539358, "Ellipsoidal organelles dubbed 'Ward bodies' contain membranous stacks; they have been observed in electron micrographs but are of unknown function". It wasn't clear if Ward bodies are in fact the same structures; the term is admittedly not in wide use. Still, if they are the same, this term should be considered as an alternative or at least mentioned in the text.

– Abstract. (1) "filled with cuticle". Is this correct or are just half of the folds- those open to the apical surface and therefore contiguous with the cuticle – filled with cuticle material. (2) "Meisosomes are therefore an essential component of the skin". Although the data is very suggestive, have the authors really provided direct evidence that this is the case? For example, with a mutation that specifically abolishes meisosomes without affecting cuticle composition for example? If not, consider using terms such as "indicate" or "strongly suggest" etc. Note that on page 10 the authors do hedge their conclusions by stating, "these novel phenotypes suggest that the meisosomes may play an important role in attaching the cuticle to the underlying epidermal cell", which seems correct. (3) "They could also be involved in relaying tensile information". To me this seems like speculation and therefore better left out of an abstract (but a fine discussion point).

– Page 4. (1) "…meisosomes across the ca. 400 available…". Suggest changing "ca." to "approximately", which is more commonly used. (2) It seems that the most compelling functional role for meiosomes (though not quite definitive) is that they are involved in epidermal-cuticle attachment. If so, perhaps more discussion about cuticle attachment would be appropriate in this section. Along those lines, the mention of a "damage sensor" seems less relevant, as this wasn't really addressed in the paper. Perhaps better left to the discussion? (3) "This mutant analysis revealed an essential role for meisosomes in maintaining the structural integrity of the cuticle and the epidermis". This seems like an overstatement as mentioned above regarding the abstract.

– Figure 1, S1, page 5. (1) It would be helpful to structurally diagram and better define the terms inner, outer, cytoplasmic, and cuticular with respect to meisosomes. Since most readers will not be familiar with the worm epidermis, the more information provided earlier the better. Although this becomes somewhat clearer in Figure 2 and later, a diagram in Figure 1 would still be useful. Also, please be consistent with inner/outer versus cytoplasmic/cuticle. Both terms are used but it will be clearer if one or the other is chosen. (2) Some of the red arrows in Figure 1 S1(B-K) don't seem to be pointing directly at the intended structures (meisosomes).

– Figure 1 and page 5. (1) "Meisosomes were frequently found in close proximity to mitochondria (85%, n=355) (Figures 1A-C). On their apical side, some folds were found close to the furrow (Figure 1C)." It wasn't clear what "close proximity" or "some" or "close to" mean exactly. Consider better defining these terms. Is there a way to quantify proximity (or lack thereof) to structures such as mitochondria or MVBs? (2) Change "Figures 1E" to "Figure 1E".

– Figure 2. Here too it would be useful to have a simple cartoon diagram what's going on as the tomography is somewhat difficult to discern. (The video is fantastic!) In addition, the TEM appears to suggest some closed or arch-like structures adject to the cuticle for some of the folds that are likely open to the cuticle. Although I expect that this is an 'artifact' of the sectioning, it might be useful to state this, and (again) a diagram could make it very clear that the inside of the folds is alternatively contiguous with either the cytoplasm or the cuticle.

– Figure 3, pages 6-7. (1) It is unclear if VHA-5::GFP provides a "bona fide" marker for meisosomes, although it may. Some assurance is based on the relevant papers (Hyenne 2015 and Liegeois 2006), however, these should be referenced at the end of the sentence stating that VHA-5 and RAL-1 are the "only known markers". (2) Despite VHA-5 likely being a marker for meisosomes, were any experiments conducting with a RAL-1 reporter and did they show similar effects (e.g., with furrow mutants etc.)? (3) I was confused by the phrase, "This indicates that MVB are not detectable with our standard fluorescence microscopy techniques." MVBs are quite easily detectable with markers such as HGRS^-1^, so perhaps the authors meant something else?

– Figure 3 S1 etc. You might want to indicate here (and throughout more explicitly) that sta-1(RNAi) is serving as a negative control, possibly just referring to it as "control RNAi" in the figures and stating that "sta-1(RNAi) was used as a negative control" in the methods section would be appropriate. You might also want to state why this was chosen as a negative control in the methods. Note that Figure 5 simply refers to sta-1(RNAi) as "control" – so please be consistent throughout.

– Page 7. "As meisosomes connect the cuticle to the epidermis". As mentioned above, is there truly direct evidence that meisosomes connect the cuticle to the epidermis? Their morphological disruption certainly corresponds with this defect, but the method of disruption (loss of certain cuticular collagens) could also potentially explain for this defect without invoking meiosomes. In any case, I would suggest using terms like "suggest" or "is consistent".

– Page 7, Figure 4, etc. I think the current description of the different classes of collagens (furrow versus annuli) will confuse some readers. "Different classes of cuticular collagen mutants exist that affect either furrows or annuli". But don't furrowless mutants like dpy-7 affect (lack) both furrows and annuli? What about stating that there are two general classes based on expression… that loss of the furrow class leads to the loss/disorganization of furrows and annuli whereas loss of the annuli class leads to alterations in furrow/annuli width or spacing.

– Figure 4 etc. (1) The measure of VHA-5/meisosome density could be better defined in the text or figure legend. Is it the amount of VHA-5 positive pixels/area or the number of discrete VHA-5 particles per area? Overall, it appears that because of the size reduction, the total VHA-5 signal per area is reduced, although a total intensity measurement or positive pixels/per area would likely demonstrate definitively and would be informative. For example, is the total area taken up by meisosomes altered in collagen mutants? (2) Is there a reason not to include p-values (***) in all figures (e.g., Figure 4S1 A-C)?

Page 8. Consider changing: (1) "…actin or microtubule…" to "…actin or microtubules…"; (2) "Since the size of meisosomes in adult worms requires furrow collagens, we then examined…" to "Since the correct size and morphology (?) of meisosomes in adult worms requires furrow collagens, we next examined…".

Figure 7, page 9. (1) Apparent mistake in Figure 7 legend title. (2) Based on the provided image in 7A, the sta-1 control seems to have wider spaced furrows than some of the other genes, such as vha-19. However, this does not appear to be the case when looking at examples in 7S. Was this quantified or might there be a better image for sta-1 in 7A if this is not the case? (3) Although the DPY-7::GFP puncta could represent material being exocytosed, Miao and Wang showed recently that DPY-7 is endocytosed during the molting cycle, so could these also be endosomes?

Figure 8, page 10, Table S1. (1) There were statements about meisosomes having fewer folds and irregular shapes in the text, but this wasn't clear from Table S1. Also, reference Table S2? (2) How often was cuticle detachment detected in mutants (vs WT). Although I don't question the truth of the statements, given that these studies are inherently descriptive (not a bad thing), any efforts to put numbers where possible on phenotypes would be useful. Even just something like "15/20 worms/images versus "0/20 in control" would be useful. (4) Indicate pm in panel 8D?

[Editors’ note: further revisions were suggested prior to acceptance, as described below.]

Thank you for resubmitting your work entitled "Meisosomes, folded membrane platforms, link the epidermis to the cuticle in *C. elegans*" for further consideration by *eLife*. Your revised article has been evaluated by Piali Sengupta (Senior Editor) and a Reviewing Editor.

The manuscript has been improved but there are some remaining issues that need to be addressed, as outlined below:

All three reviewers found that the manuscript has been improved, but there are some inappropriate statements and overinterpretations. Please revise the text accordingly.

*Reviewer #1 (Recommendations for the authors):*

This revised manuscript made a much-improved investigation of the meisosome structures and their biogenesis process compared with the previous version. However, the most important question – "what are the major functions of meisosomes?" was still not satisfactorily answered. Like the previous version, functional studies were performed on mutants of collagen molecules that are not even localized to the meisosomes, therefore can only reflect functions of collagens but not of meisosomes. For example, the data showed that those collagen mutants exhibit lateral epidermal detachment, as well as smaller meisosomes. But no effort was put into sorting out the cause-effect relationship between these two phenotypes. The authors claim that "the meisosomes may play an important role in attaching the cuticle to the underlying epidermal cell". Yet there is no evidence supporting that epidermal detachment is caused by defective meisosomes. It is equally possible that it's the detached lateral epidermis that caused meisosome malformation. In a word, this study did an excellent job analyzing the meisosome structures and collagen mutants but did not contribute much to our understanding of meisosome functions.

*Reviewer #2 (Recommendations for the authors):*

The authors have done many works to revise the manuscript. Most of my previous questions have been addressed. Given the well-characterization of the structure using the ultrastructural approach, it looks promising now and would be interesting to the worm community. However, as the function of this stack is still unclear, there are still a few issues.

1. I did not see the specific reason why this apical membrane stack has to be renamed as "meisosome". The authors claimed that "Given their superficial similarity, we refer to these structures as meisosomes, for multifold-eisosomes." which obviously is not a good reason. The function of Meisosome is unknown, nor is the function similar to eisosome. There is no yeast eisosome homology protein (e.g. Pli-1) has been found specifically localized to this structure.

This structure has been described as "Ward bodies" or "Apical membrane Stack", it would be more suitable to continue to use these names for appreciating previous studies. It would be even better to abbreviate "Apical Membrane Stack" as "AMStack".

Besides, the paper PMC6214159, introduces the history of the eisosome was named. "Early functional studies in budding yeast suggested that eisosomes might represent sites of endocytosis. Hence, the structures were named eisosomes after the Greek words eis, meaning entry, and soma, meaning body. Subsequent studies in budding yeast and other organisms have shown that eisosomes are not sites of endocytosis, but the name has remained." It would be confusing if meisosomes will still be used to describe the structure.

2. There are many citation issues, for example:

As Wood 1988, White et al., 1986, and Liegeois et al., 2006 have described the structure, I would strongly suggest introducing the structure by citing these papers in the introduction session.

In the discussion, "Notably, a recent study reported the isolation of mutants with an abnormal pattern of VHA-5::RFP in the 319 epidermis but attributed this to a change in MVBs (Shi et al., 2022)" The paper Shi et al., 2022 is missing in the references.

"Since previous studies (Liegeois et al., 2006), and the results presented here, show that VHA-5 is predominantly a marker of meisosomes, more so than of MVBs, we hypothesise that the one gene that Shi et al. characterised in detail, fln-2, which encodes the F-actin cross-linking protein filamin (Zhao et al., 2019), could actually be involved in the formation and/or maintenance of meisosomes." What is Shi et al? Zhao et al., 2019 are also missing in the references.

Besides, it is not clear why fln-2 will be involved in the formation and/or maintenance of this structure.

4. Overstatements

For example, In the abstract: "As meisosomes co-localise to macrodomains enriched in phosphatidylinositol (4,5) bisphosphate, they might act, like eisosomes, as signalling platforms, to relay tensile information from the aECM to the underlying epidermis, as part of an integrated stress response to damage." and discussion: "So, it is tempting to propose that analogous to eisosomes, meisosomes could have a role as a signalling platform in response to stress." I did not see the logic that the authors claim this structure could function in response to stress/damage.

4. Another issue, to me, it looks like Figure 3I and 3J are the same as Figure 3S2G and 3S2H, only with different contrast.

*Reviewer #3 (Recommendations for the authors):*

I strongly support the publication of this manuscript in eLife. The authors have addressed previous concerns in a reasonable way and the paper contains a great deal of high-quality interesting data that should be of general interest. The paper is also exceptionally well presented and written and I very much enjoyed reading it.

---

## [Author Response]

[Editors’ note: The authors appealed the original decision. What follows is the authors’ response to the first round of review.]

Comments to the Authors:We are sorry to say that, after consultation with the reviewers, we have decided that this work will not be considered further for publication by eLife.Specifically, the reviewers have raised concerns on whether VHA-5::GFP can serve as a "bona fide" meisosome marker and whether meisosomes serve as attachment platforms between the cuticle and the epidermis.

Regarding the first part, we apologise if were not sufficiently clear. Previous work from the Labouesse group, with whom we have collaborated for two decades, already strongly suggested that VHA-5::FP can indeed serve as a "bona fide" meisosome marker. Thus in the paper we cite, Liegeois et al., 2006, there is a qualification of immunogold staining that shows that >85% of VHA-5 is found in meisosomes, as well as a comparison between immunofluorescence with a VHA-5 antiserum and the fluorescence of VHA-5::RFP, leading the authors to state (Figure S5D), “Note that VHA-5 is very strongly enriched at the apical membrane stacks compared with WRT-2::GFP [principally in MVBs], which may correspond to the dotted structures observed via VHA-5::mRFP fluorescence [Figure 1 E] and by immunofluorescence [Figure 1 F]”. Our initial aim was to demonstrate that the particular reporter system that we used was an appropriate tool for our studies, and recapitulated the findings of Liegeois et al., given the known artefacts that can arise when using chimeric reporter proteins, as highlighted by the reviewers. In order to dispel any concern, we now provide a considerable amount of new data, using both a single-copy insertion and direct genome engineering to knock *GFP* into the VHA-5 locus. In all cases, we obtain qualitative and quantitative data that matches that presented previously. To take this one step further, we also now provide CLEM data that unequivocally supports the contention that VHA-5::GFP can indeed serve as a "bona fide" meisosome marker. Further, we now show that VHA-5::mRFP co-localises with two different plasma membrane markers (PH-PLCδ and CAAX).

We note that this issue has taken on a new relevance given the publication of a vey recent study (Shi *et al.,* 2022, https://doi.org/10.1083/jcb.202201020). Here, in direct contrast to the Liegeois study, and our findings, the authors take VHA-5::RFP as an MVB marker. Surprisingly, Shi *et al.*’s interpretation is even at odds with their own observations, since they also see no co-localisation with HGRS-1, the well characterised HGR orthologue that marks MVB biogenesis. We mention this discrepancy in the discussion.

With regards the question of whether meisosomes serve as attachment platforms between the cuticle and the epidermis, as we now explain, there are no known treatments that affect specifically meisosomes, nor mutants that have been demonstrated to affect directly their integrity. The recent paper mentioned above does identify a number of possible candidates, but at this stage, given the confusion regarding their interpretation, pursuing the role of these candidates is beyond the scope of the current study. Having said this, we have taken advantage of our observation that meisosomes are disrupted in Dpy furrow mutants to characterise more precisely the structural consequences. We now show by classical TEM of more Dpy furrow mutants and by SBF of entire transversal sections, that it is precisely above the areas containing meisosomes where the basal cuticle is normally anchored into the apical epidermis that one observes detachment, while the cuticle above the muscle regions retains its usual close apposition. We further illustrate the structural complementarity and mutual exclusion of meisosomes and hemidesmosomes in these epidermal regions using a strain expressing both MUP-4::GFP and VHA-5::RFP (Figure 3D). We also have unpublished evidence that hemidesmosome are still present in furrow Dpy mutants, as reported for *dpy-2(e8)*, *dpy-7(e88)*, *dpy-10(e128)* in Wang et al., 2020 (doi:10.1242/jcs.246793). So, if it is clear that hemidesmosomes serve to anchor the muscles to the cuticle through the epidermis, we believe that it is reasonable to propose that in the main lateral epidermis and dorso-ventral ridges, the meisosomes contribute to the attachment of the epidermis to the cuticle.

We address outstanding points from the individual reviewer comments below; any point that is omitted has been addressed above, or in the response to a similar comment from another reviewer. We have changed the text accordingly and modified the organisation to have a better flow.

Reviewer #1 (Recommendations for the authors):1. Transgenes carrying fluorescent fusion reporters, especially multi-copy extrachromosomal arrays (such as the frEx624[pML670(VHA-5::GFP, unc-122p::GFP)] used in most figures of this manuscript), are known to misbehave and often do not faithfully reflect the endogenous distribution patterns of the protein-of-interest. Therefore, at least one alternative approach for meisosome morphological analysis should be provided to confirm the key findings of this paper. There are indeed a few TEM images of the furrow mutants showing one or two smaller meisosome structures (Figure 8), which is very good itself. Unfortunately there are no parallel-performed controls in this dataset and no quantification analysis.

In addition to explaining how prior work has established that the pattern of VHA-5::FP reflects that of the endogenous protein (see above), and the use of the new knock-in and single insertion reporters (new Figures 4,5 and 6 and quantification in Figure 6 source data file), we also now make clearer the fact that the TEM controls are provided by both the wild-type worms in Figure 1, and by the non furrow collagen mutant *dpy-13* (Figure 7) and that the differences are quantified in Table 1 and Table 1 source data file (previously Table S1).

2. In most RNAi experiments, knockdown of the STAT family transcription factor sta-1 was used as negative control instead of the commonly used empty RNAi clone vector L4440. However, no explanation was provided as for why this particular negative control was chosen. In fact, the gene name sta-1 did not appear at all in the entire maintext. The authors need to explain why they assume that sta-1 deficiency does not affect the epidermis and the cuticle (even indirectly, because sta-1 is expressed in the body-wall muscles adjacent to the epidermis, and defects in the muscles can greatly affect epidermal biogenesis and function), and what are the advantages of choosing sta-1 RNAi as the negative control over the empty RNAi vector or RNAi clones targeting other genes.

The reviewer questioned the pertinence of relying solely on *sta-1* as a control for RNAi experiments. There are 2 aspects to our response. The first is that since we have used *sta-1(RNAi)* in literally thousands of experiments under a broad range of conditions, we know that it does not affect the key aspects of worm physiology or gene expression in the epidermis. To give one example, while even a mild decrease in the expression of furrow collagens leads to the expression of *nlp-29*, *sta-1(RNAi)* does not (original Figure 6S1). We now make this clearer in the revised manuscript. But more importantly, for all of the relevant experiments, we use a battery of RNAi controls: non-furrow *dpy* genes that provoke a similar Dpy phenotype. This allows us to determine that change *per se* of the worm size and morphology, that are seen in all Dpy mutants, do not cause the fragmentation of the meisosomes. We have now made these controls clearer by using a colour code explained in the schema in Figure 6A (green for annuli Dpy mutants and blue for furrow Dpy mutants). We also provide the quantification of the size of the worms upon RNAi inactivation along the induction of the *nlp-29*p::GFP reporter (Figure 6S1) and explain in the Materials and methods that the inactivation of the Dpy collagen genes in all our RNAi experiments was validated by checking that the worms were phenotypically Dpy (short size and fat), and that only the furrow Dpy collagen inactivation lead to induction of *nlp-29p::GFP* reporter, as per our previous studies (Dodd et al., 2018; Pujol, Zugasti, et al., 2008; Zugasti et al., 2014).

Reviewer #2 (Recommendations for the authors):1. Apical membrane stack is well known and much better to describe this folded structure, not necessary to use "Meisosomes" to rename it. And this structure should not be defined as an organelle.

Even if a function is still elusive, we believe that meisosomes conform to the definition of “A differentiated structure within a cell, such as a mitochondrion, vacuole, or chloroplast, that performs a specific function” and prefer to use the term, which Reviewer#3 used too.

2. VHA-5 is well known to localize to the apical membrane stacks (Liegeois 2006) and could be served as marker of apical membrane structure. How does VHA-5::GFP expression, puncta size, and intensity correlate with the structure of the apical membrane stack? The information about the VHA-5::GFP transgene frEx624[pML670(VHA-5::GFP, unc-122p::GFP)] is not very clear. Is VHA-5::GFP expression under its own promoter or another promoter? It is known that the extrachromosomal array of transgene show various expression level, thus the VHA-5::GFP knock-in strain should be used to quantify the localization, puncta size, and intensity. With the CRISPR-Cas9 genome editing method, it should be easy to make knock-in strain now.

As explained above, and as presented in original Figure 4S1, for all the RNAi experiments, tests were also done in parallel to assay (i) the size and (ii) the level of induction of the *nlp-29p::GFP* reporter, (see Figure 6S1 and Figure 6 source data file). We have tried to make this clearer in the revised manuscript in the main text, figure legends and Materials and methods.

3. What's the function of apical membrane stack in collagen secretion?

As we state in the text, “VHA-5 has been shown to have an essential role in alae formation and secretion of Hedgehog-related peptides through exocytosis via MVBs, but not to be involved in secretion of the collagen DPY-7, nor in meisosome morphology (Liegeois et al., 2006)”. There is therefore no reason to believe meisosomes to be involved in collagen secretion, especially as they have no apparent direct connection with Golgi, ER or other cytoplasmic vesicles. If deemed appropriate, we could expand on this further in the manuscript.

Reviewer #3 (Recommendations for the authors):Comments are in (approximate) relative order:– The term meisosomes. The authors have (re)named the folded membrane structures "meisosomes". These were previously referred to as "apical membrane stacks" by Labouesse and colleagues, although the term "meisosomes" is perhaps more efficient and descriptive. Another term that may be relevant is "Ward bodies". According to PMID: 23539358, "Ellipsoidal organelles dubbed 'Ward bodies' contain membranous stacks; they have been observed in electron micrographs but are of unknown function". It wasn't clear if Ward bodies are in fact the same structures; the term is admittedly not in wide use. Still, if they are the same, this term should be considered as an alternative or at least mentioned in the text.

There is confusion in the worm literature about Ward bodies, as no detailed characterization has been published since the original observations reported in the first worm book, Wood et al., 1988. In PMID: 23539358, the authors differentiate between membrane stacks and Ward bodies, the latter being cytoplasmic vesicular components, based on the same worm book reference. We prefer not to cite Ward's bodies, so as not to create more confusion.

– Figure 1 and page 5. (1) "Meisosomes were frequently found in close proximity to mitochondria (85%, n=355) (Figures 1A-C). On their apical side, some folds were found close to the furrow (Figure 1C)." It wasn't clear what "close proximity" or "some" or "close to" mean exactly. Consider better defining these terms. Is there a way to quantify proximity (or lack thereof) to structures such as mitochondria or MVBs

This is a very interesting question, seen the recent literature on mitochondria and plasma membrane interactions (as in PMID: 34481840). We mention in the result that 85 % of meisosome are in close proximity to mitochondria. In tomography or TEM, some meisosomes are almost touching mitochondria, but we could still see that their membranes remained separated (Figure 2). More quantitative statistical analyses of the proximity and its functional relevance would require an in-depth study, which we consider beyond the scope of this paper.

(2) Despite VHA-5 likely being a marker for meisosomes, were any experiments conducting with a RAL-1 reporter and did they show similar effects (e.g., with furrow mutants etc.)?

In our hands, the previously characterised RAL-1::YFP reporter (kindly provided by Jeremy Nance) did not give a signal of sufficient intensity in the epidermis to permit correct analyses. We could mention this if it is considered appropriate.

(3) I was confused by the phrase, "This indicates that MVB are not detectable with our standard fluorescence microscopy techniques." MVBs are quite easily detectable with markers such as HGRS^-1^, so perhaps the authors meant something else?

We agree and the literature using VHA-5 markers is very confusing. The majority of the fluorescent signal from VHA-5::FP in the epidermis comes from protein associated with meisosomes. As explained above, only a small fraction of the protein is associated with MVBs (15%, see Liegois et al. 2016, Figure S5). In agreement with the recent Shi et al. paper, we observed no colocalization between VHA-5::FP and HGRS^-1^::FP. Whereas Shi et al. conclude that “HGRS^-1^ puncta do not overlap with FLN-2– or VHA-5–positive vesicles in the epidermis, which suggests that HGRS^-1^ and FLN-2 or VHA-5 are enriched on different vesicular compartments or subregions”, since HGRS^-1^ is undoubtedly a MVB marker, we propose a more parsimonious explanation and have modified the text accordingly.

Figure 8, page 10, Table S1. (1) There were statements about meisosomes having fewer folds and irregular shapes in the text, but this wasn't clear from Table S1. Also, reference Table S2? (2) How often was cuticle detachment detected in mutants (vs WT). Although I don't question the truth of the statements, given that these studies are inherently descriptive (not a bad thing), any efforts to put numbers where possible on phenotypes would be useful. Even just something like "15/20 worms/images versus "0/20 in control" would be useful. (4) Indicate pm in panel 8D?

We now make a clearer reference to Table 1 for TEM analysis of meisosome length, and provide the source data file with the raw data. The number of worms analysed is mentioned in the figure legends, and we provide the Figure 6-Source data file with the quantification and statistics after segmentation of VHA-5 objects, including the number of worms and the total surface area analysed per condition, and this using different VHA-5 reporter strains in wild type and different collagen RNA inactivation or mutants. We also now present the cuticle detachment phenotype through a TEM analysis of several different furrow mutants in the new Figure 7 and 7S1. We also have scanned the cuticle of wildtype and *dpy-2* mutants using SBF, where entire transversal sections were acquired over a length of 21.5 for a wild-type and 34.4 µm for a *dpy-2* mutant worm, revealing several sites of detachments over most of the surface of the lateral epidermis or/and dorsal and ventral ridges only in the *dpy-2* mutant (see new Figure 7H). Moreover, using either the CAAX or the PH domain fluorescent markers in the background of a red fluorescent cuticular collagen (ROL-6::mScarlet, provided by Cathy Savage-Dunn, a co-author of the revised manuscript), we show that there is an abnormal presence of plasma membrane material in the cuticle (now presented and quantify in Figure 8B), corroborating the TEM observation of vesicular and cytoplasmic extrusions in furrow mutants (presented in figure 8A, with pseudo coloured epidermal cell to highlight the plasma membrane).

[Editors’ note: what follows is the authors’ response to the second round of review.]

Reviewer #1 (Recommendations for the authors):This revised manuscript made a much-improved investigation of the meisosome structures and their biogenesis process compared with the previous version. However, the most important question – "what are the major functions of meisosomes?" was still not satisfactorily answered. Like the previous version, functional studies were performed on mutants of collagen molecules that are not even localized to the meisosomes, therefore can only reflect functions of collagens but not of meisosomes. For example, the data showed that those collagen mutants exhibit lateral epidermal detachment, as well as smaller meisosomes. But no effort was put into sorting out the cause-effect relationship between these two phenotypes. The authors claim that "the meisosomes may play an important role in attaching the cuticle to the underlying epidermal cell". Yet there is no evidence supporting that epidermal detachment is caused by defective meisosomes. It is equally possible that it's the detached lateral epidermis that caused meisosome malformation. In a word, this study did an excellent job analyzing the meisosome structures and collagen mutants but did not contribute much to our understanding of meisosome functions.

We were pleased that Reviewer #1 found the manuscript “much-improved”. We believe that we explained fully in our previous rebuttal the reasons for which we have not been able to provide more solid insight into the function of meisosomes (in a nutshell, because of the lack of mutants only altering meisosome structure). To make this clearer, we have altered the sentence, “Further study will be required to determine the catalogue of proteins that are needed for meisosome formation and maintenance.” to “Further study will be required to determine the catalogue of proteins that are needed for meisosome formation and maintenance. This would then allow the function of meisosomes to be addressed directly.”.

Reviewer #1 also wrote, “The authors claim that "the meisosomes may play an important role in attaching the cuticle to the underlying epidermal cell". Yet there is no evidence supporting that epidermal detachment is caused by defective meisosomes. It is equally possible that it's the detached lateral epidermis that caused meisosome malformation”. We were careful to modulate the text in the revised version. This is the reason we wrote, “may play”, as it is a hypothesis. Indeed, the complete sentence is, “Together, these phenotypes suggest that the meisosomes may play an important role in attaching the cuticle to the underlying epidermal cell”. Nevertheless, to address the reviewer’s concern, we have further qualified the text: “Together, these phenotypes lead us to hypothesise that the meisosomes may play an important role in attaching the cuticle to the underlying epidermal cell”, and have added “Although this fragmentation could be a consequence of the detachment” to the Discussion. Perhaps more importantly, to assuage this and the other reviewers’ reserves, we also propose to change the title to “Meisosomes, folded membrane microdomains between the apical extracellular matrix and epidermis”.

Reviewer #2 (Recommendations for the authors):The authors have done many works to revise the manuscript. Most of my previous questions have been addressed. Given the well-characterization of the structure using the ultrastructural approach, it looks promising now and would be interesting to the worm community. However, as the function of this stack is still unclear, there are still a few issues.1. I did not see the specific reason why this apical membrane stack has to be renamed as "meisosome". The authors claimed that "Given their superficial similarity, we refer to these structures as meisosomes, for multifold-eisosomes." which obviously is not a good reason. The function of Meisosome is unknown, nor is the function similar to eisosome. There is no yeast eisosome homology protein (e.g. Pli-1) has been found specifically localized to this structure.This structure has been described as "Ward bodies" or "Apical membrane Stack", it would be more suitable to continue to use these names for appreciating previous studies. It would be even better to abbreviate "Apical Membrane Stack" as "AMStack".Besides, the paper PMC6214159, introduces the history of the eisosome was named. "Early functional studies in budding yeast suggested that eisosomes might represent sites of endocytosis. Hence, the structures were named eisosomes after the Greek words eis, meaning entry, and soma, meaning body. Subsequent studies in budding yeast and other organisms have shown that eisosomes are not sites of endocytosis, but the name has remained." It would be confusing if meisosomes will still be used to describe the structure.2. There are many citation issues, for example:As Wood 1988, White et al., 1986, and Liegeois et al., 2006 have described the structure, I would strongly suggest introducing the structure by citing these papers in the introduction session.In the discussion, "Notably, a recent study reported the isolation of mutants with an abnormal pattern of VHA-5::RFP in the 319 epidermis but attributed this to a change in MVBs (Shi et al., 2022)" The paper Shi et al., 2022 is missing in the references."Since previous studies (Liegeois et al., 2006), and the results presented here, show that VHA-5 is predominantly a marker of meisosomes, more so than of MVBs, we hypothesise that the one gene that Shi et al. characterised in detail, fln-2, which encodes the F-actin cross-linking protein filamin (Zhao et al., 2019), could actually be involved in the formation and/or maintenance of meisosomes." What is Shi et al? Zhao et al., 2019 are also missing in the references.Besides, it is not clear why fln-2 will be involved in the formation and/or maintenance of this structure.4. OverstatementsFor example, In the abstract: "As meisosomes co-localise to macrodomains enriched in phosphatidylinositol (4,5) bisphosphate, they might act, like eisosomes, as signalling platforms, to relay tensile information from the aECM to the underlying epidermis, as part of an integrated stress response to damage." and discussion: "So, it is tempting to propose that analogous to eisosomes, meisosomes could have a role as a signalling platform in response to stress." I did not see the logic that the authors claim this structure could function in response to stress/damage.4. Another issue, to me, it looks like Figure 3I and 3J are the same as Figure 3S2G and 3S2H, only with different contrast.

We were also pleased to read Reviewer #2’s comments, “Most of my previous questions have been addressed. Given the well-characterization of the structure using the ultrastructural approach, it looks promising now and would be interesting to the worm community”. Reviewer #2 returned to the question of a name. Firstly, we should point out, that “Ward bodies” is not suitable. The term was first used in the anatomy chapter by J. White in the “green” book (Wood et al. 1988), and, as mentioned in our previous response, refers to an internal structure that is distinct from “apical plasma membrane folded and stacked”: see Figure 2aandc p85 (Wood et al. 1988), and Figure 3 in the 2012 review by Chisholm et al. (PMID: 23539358). In Liégois et al. 2006, the authors refer to the structure, that were not their primary focus, as “apical membrane stacks” and not “Ward bodies”. Secondly, as we wrote previously, “meisosome” is a neologism that has been warmly received by many colleagues in the field. Just like seahorse or catfish, “meisosomes” indicates a structure that looks like multiple eisosomes. We believe that the fact that “eisosome” itself is a misnomer is not a relevant consideration, since eisosome is now widely used in the fungi community and is very unlikely to change. We note that the other 2 reviewers have adopted meisosome. We believe that it will avoid the emergence of another acronym like “AMS”, if we continue to call this structure ‘apical membrane stack’.

Otherwise, we are grateful to the reviewer for pointing out the problems with the bibliography that have now been corrected. We had chosen to present only part of the CLEM data in the main Figure 3, and present the full set in Figure 3S2. This explains the apparent duplication of the panels 3I-H, as explained in the legend to Figure 3 (“see associated Figure 3—figure supplement 2 for the detailed procedure”). And to answer, “it is not clear why *fln-2* will be involved in the formation and/or maintenance of this structure”, we have changed the text to “Since previous studies (Liegeois et al., 2006), and the results presented here, show that VHA-5 is predominantly a marker of meisosomes, more so than of MVBs, these mutants with an abnormal pattern of VHA 5::RFP more likely affect meisosomes.”

The reviewer also wrote, “I did not see the logic that the authors claim this structure could function in response to stress/damage”, quoting our text, "As meisosomes co-localise to macrodomains enriched in phosphatidylinositol (4,5) bisphosphate, they might act, like eisosomes, as signalling platforms, to relay tensile information from the aECM to the underlying epidermis, as part of an integrated stress response to damage." and discussion: "So, it is tempting to propose that analogous to eisosomes, meisosomes could have a role as a signalling platform in response to stress." We have altered the former to, “As meisosomes co-localise to macrodomains enriched in phosphatidylinositol (4,5) bisphosphate, they could conceivably act, like eisosomes…”. For the latter, we are not sure how to further qualify, “it is tempting to propose”, as this is already so far removed from a “claim”. But we would of course welcome any proposition if the editors consider that the text does need to be changed. In addition, we have added a brief description of the signalling function of the eisosomes to the discussion, “They can disassemble in minutes to buffer changes in membrane tension, protecting cells from osmotic shock, but also activate membrane stress signalling pathways through the release of BAR domain containing proteins (Appadurai et al., 2020; Lemiere et al., 2021). Eisosome-like structures are conserved throughout fungi, microalgae and lichens (Lee et al., 2015; Zahumensky and Malinsky, 2019), but there are no direct orthologues for core components, such as Pil1 or LSP-1, in animals.”